# The involvement of the human prefrontal cortex in the emergence of visual awareness

**Zepeng Fang[1], Yuanyuan Dang[2], Zhipei Ling[2], Yongzheng Han[3], Hulin Zhao[2]\*, Xin Xu[2]\*, Mingsha Zhang[1]\***

[1]State Key Laboratory of Cognitive Neuroscience and Learning and IDG/McGovern Institute for Brain Research, Division of Psychology, Beijing Normal University, Beijing, China; [2]Department of Neurosurgery, Chinese PLA General Hospital, Beijing, China; [3]Department of Anesthesiology, Peking University Third Hospital, Beijing, China

**\*For correspondence:**
zhaohulin_90@sohu.com (HZ);
xuxinmm@hotmail.com (XX);
mingsha.zhang@bnu.edu.cn (MZ)

**Competing interest:** The authors declare that no competing interests exist.

**Abstract** Exploring the neural mechanisms of awareness is a fundamental task of cognitive neuroscience. There is an ongoing dispute regarding the role of the prefrontal cortex (PFC) in the emergence of awareness, which is partially raised by the confound between report- and awareness-related activity. To address this problem, we designed a visual awareness task that can minimize report-related motor confounding. Our results show that saccadic latency is significantly shorter in the aware trials than in the unaware trials. Local field potential (LFP) data from six patients consistently show early (200–300ms) awareness-related activity in the PFC, including event-related potential and high-gamma activity. Moreover, the awareness state can be reliably decoded by the neural activity in the PFC since the early stage, and the neural pattern is dynamically changed rather than being stable during the representation of awareness. Furthermore, the enhancement of dynamic functional connectivity, through the phase modulation at low frequency, between the PFC and other brain regions in the early stage of the awareness trials may explain the mechanism of conscious access. These results indicate that the PFC is critically involved in the emergence of awareness.

## eLife assessment

This paper reports **valuable** results regarding the potential role and time course of the prefrontal cortex in conscious perception. Although the sample size is small, the results are **convincing**, and strengths include the use of several complementary analysis methods. The behavioral test includes subject report such that the study does not allow for distinguishing between (phenomenal) awareness and conscious access; nevertheless, results do advance our understanding of the contribution of prefrontal cortex to conscious perception.

## Introduction

A conscious state means that a person has some kind of subjective experiences, such as seeing a painting, hearing a sound, thinking about a problem, or feeling an emotion (*Koch et al., 2016*). The neural mechanism of consciousness is one of the most fundamental, exciting and challenging pursuits in the 21st-century brain science (*Mashour, 2018*). Among the various aspects of consciousness, visual awareness, as a core of consciousness, has attracted enormous attention in consciousness research. Visual awareness seems quite simple: we see an object and seem to immediately know its shape, color

and other properties. However, the neural processes behind the emergence of visual awareness are quite sophisticated and largely unknown (*Dehaene et al., 2017*).

The typical method to study the neural mechanism of visual awareness is to present external stimuli with similar physical properties but to elicit different conscious experiences, such as using near perceptual threshold stimuli, visual masking, or attentional blinks (*Kim and Blake, 2005*; *Dehaene and Changeux, 2011*). Then, the aware and unaware states are determined through the subjective reports of participants, and the awareness-related neural activity is dissociated by the brain activation difference between the two states. Using this approach, some studies have found that activities in broad brain regions are correlated with visual awareness. For example, an fMRI study of lexical visual masking found that only consciously perceived and reported words can induce activation of higher-order cortical regions such as the frontal and parietal lobes (*Dehaene et al., 2001*). Another study using a visual masking task found changes in the amplitude of intracranial electroencephalography (iEEG) signals and high-gamma power in occipital-temporal regions, which indicated that these regions played an important role in the emergence of awareness (*Fisch et al., 2009*).

However, there is still much debate about the origin of conscious awareness in the brain. Some hypotheses (global neuronal workspace theory, higher-order theory, etc.) argued that visual awareness arose from the frontoparietal network, in which the prefrontal cortex played a key role, and they argued that the earliest biomarker of awareness in EEG is the P3b wave onset approximately 300ms after stimulus onset (*Mashour et al., 2020*; *Brown et al., 2019*). For example, based on the electrophysiological study of a visual awareness-guided saccade selection paradigm in macaques, by comparing the action potential firing characteristics of neurons in the dorsolateral prefrontal cortex (dlPFC), V4 and V1, only dlPFC neurons were closely associated with visual awareness states. Therefore, the authors argued that stimuli must elicit a threshold level of prefrontal activity for conscious detection to occur (*van Vugt et al., 2018*). In contrast, other hypotheses (integrated information theory, recurrent processing theory, etc.) proposed that the 'hot zone' of the posterior cortex of the brain, including the parietal lobe, occipital lobe, and temporal lobe, is sufficient to generate visual awareness, and its biomarkers on EEG are the relatively earlier visual awareness negativity (VAN) at approximately 200ms after stimulus onset (*Koch et al., 2016*; *Koch, 2018*; *Lamme, 2018*; *Dembski et al., 2021*), whereas the PFC is not necessary for the emergence of conscious experience.

This controversy is partially caused by the confounding effect of report-related activity, such as motor planning and execution, on awareness-related activity. The reason is that the study of awareness usually requires subjective reports to determine different awareness states, while the prefrontal cortex is closely related to such report-related functions (*Seth and Bayne, 2022*). To solve this problem, some researchers use the no-report paradigm to rule out report-related impacts. For instance, a functional magnetic resonance imaging (fMRI) study employing human binocular rivalry paradigms found that when subjects need to manually report the changing of their awareness between conflict visual stimuli, the frontal, parietal, and occipital lobes all exhibited awareness-related activity. However, when report was not required, awareness-related activation was largely diminished in the frontal lobe but remained in the occipital and parietal lobes (*Frässle et al., 2014*). Nevertheless, the no-report paradigm may overestimate the neural correlates of awareness by including unconscious processing, because it infers the awareness state through other relevant physiological indicators, such as optokinetic nystagmus and pupil size (*Tsuchiya et al., 2015*). In the absence of subjective reports, it remains controversial regarding whether the presented stimuli are truly seen or not. Therefore, neither the typical report paradigm nor the no-report paradigm is ideal for visual awareness research, thus, it is difficult to solve the problem of whether the PFC participates in conscious perception.

In addition, when studying the role of the PFC in visual awareness, the noninvasive technologies mostly used in previous studies, that is, Functional magnetic resonance imaging (fMRI) and M/EEG have limitations. Although fMRI research can detect the whole brain activation pattern, it is difficult to describe the dynamic changes and sequences of different brain regions in the rapid process of visual awareness due to its limited time resolution (approximately several seconds). For example, even though a recent fMRI study using the no-reported paradigm found that there are still visual awareness-related activities in the PFC (*Kronemer et al., 2022*), it remains unclear when these activities begin and whether PFC activity occurs earlier or later than other brain regions. At the same time, although EEG and MEG have a very high temporal resolution, their low spatial resolution and low signal-to-noise ratio limit their interpretation of awareness-related activities, especially high-frequency activities

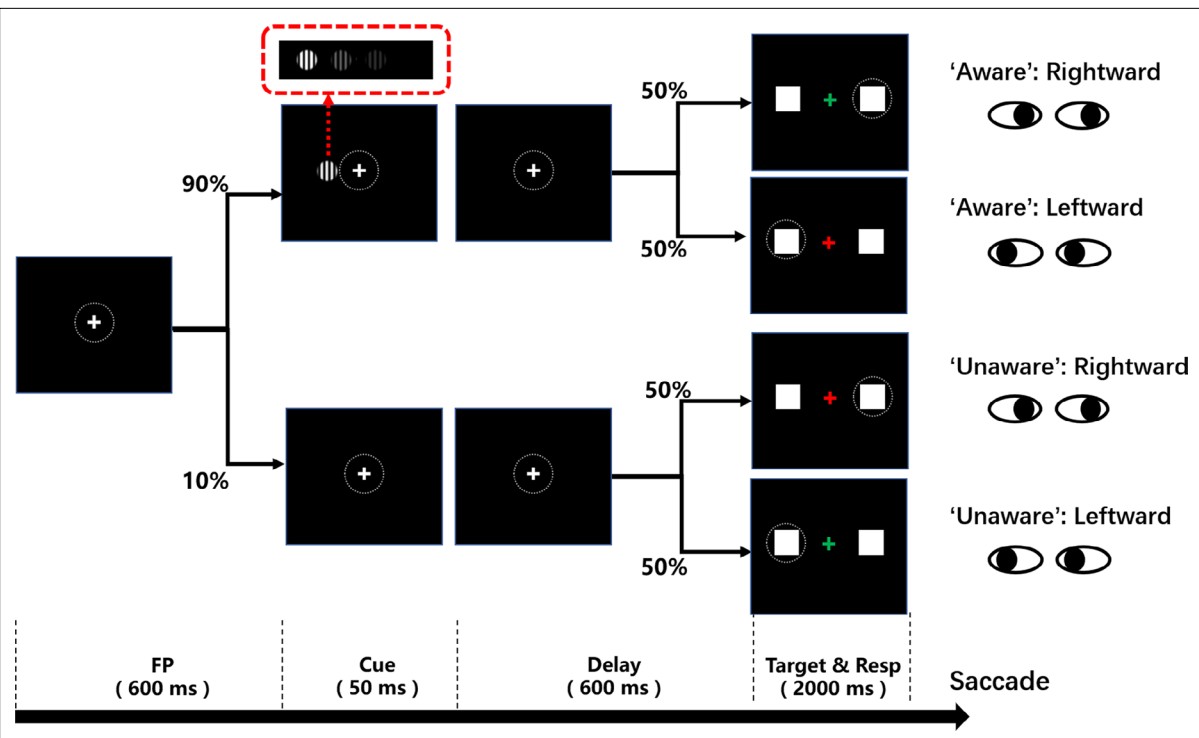

**Figure 1.** Schematic diagram of the visual awareness task. A trial started when a fixation point (0.5°x0.5°, white cross) appeared at the center of the screen (radius of eye position check window is 4°, the dotted circle). After the subject fixated on the fixation point for 600ms, a cue stimulus (Gabor grating, 2x2° circle) was presented for 50ms at a fixed position (7°) on the left (or right, see Methods) side of the screen for all participants. In 70% of the trials, the grating contrast was maintained near the subject's perceptual threshold by a staircase method; in 10% of the trials, the stimulus contrast was well above the threshold; and in the other 20% of the trials, the stimulus contrast was 0, namely, no stimulus appeared. After another 600ms delay, the color of the fixation point turned red or green, and two saccade targets (1x1°, white square) appeared at fixed positions (10°) on the left and right sides of the screen. If the grating was seen, the green fixation point was required to make a saccade to the right target, while the red fixation point was required to make a saccade to the left target. If the grating was not seen, the rule of saccadic direction was inverted.

(highly correlated with local population neuronal activities). For example, although increasing evidence shows that the earliest biomarker of visual awareness in EEG is more likely to be the VAN, which starts at approximately 180–300ms (may be delayed to 300–460ms for low-visible stimulus) and peaks at the parieto-occipital region (*Northoff and Lamme, 2020*; *Koivisto and Revonsuo, 2010*; *Koivisto et al., 2008*), rather than the P3b wave onset at approximately 300–500ms which peaks in the central frontoparietal region, it is still unclear whether there are local visual awareness-related activities in the PFC during the VAN period. In addition, a few recent studies have suggested that awareness may be related to the activities of brain networks, including the PFC, rather than that of a single brain region. However, most of these brain network studies used noninvasive research methods (*Kronemer et al., 2022*; *Huang et al., 2023*), and due to the limitations mentioned above, it is difficult to detect the dynamic changes in brain networks in a more precise spatial-temporal profile during the generation of visual awareness.

To address the questions mentioned above, we designed a novel visual awareness task (*Figure 1*) that can minimize motor-related confounding, while retaining the explicit subjective report. The contrast of the cue stimulus (grating) was roughly divided into three levels: well above the visual perceptual threshold, near the visual perceptual threshold and zero contrast (no stimulus). During each experiment, in nearly 80% of trials, the contrast of the grating was close to the subject's visual perceptual threshold so that the subjects could sometimes see the grating and sometimes could not. Subjects were required to choose a saccade direction according to the visual awareness state (seeing the grating or not seeing the grating) and the color of the fixation point. Notably, subjects were unable to choose the saccade direction or effectively prepare for saccades until the color of the fixation point was changed. Moreover, the subjects needed to choose the saccade direction, both in the awareness and unawareness states, according to the color of the fixation point so that the report

behaviors were matched between the two awareness states. Such a paradigm can effectively disso-ciate awareness-related activity from motor-related activity in terms of time (the fixation point changes color 650ms after the grating onset) and report behavior, thereby being report-independent of the awareness states (*Merten and Nieder, 2012*), thus minimizing the impact of motor-related factors on visual awareness. In addition, saccade parameters (such as reaction time, etc.) under different aware-ness states can be compared to explore the influence of visual awareness on saccade behavior. Then, we applied the above behavioral paradigm to six clinical patients with implanted electrodes (stereo-electroencephalography, sEEG) in prefrontal and other cortices and 10 healthy subjects. The behavior and local field potential (LFP) data of six patients were recorded while they performed the task, and we also collected the behavioral data of 10 healthy subjects for comparison with the patients' behav-ioral data.

## Results

### Behavioral results

To detect the effect of cue stimulus (grating) contrast on the emergence of visual awareness, we first analyzed the proportion of participants being aware of the grating as a function of its contrast. *Figure 2A* shows exemplified sessions from a patient and healthy subject, and *Figure 2B* shows the population results of the two participant groups. The results showed that with increasing grating contrast, the ratio of reporting 'awareness' of the grating gradually increased for all participants (including patients and healthy subjects), showing a classical psychometric curve. Such results showed that the grating contrast level had a direct impact on the emergence of visual awareness and thus is a reliable way to study visual awareness. In addition, in trials with no grating and grating contrast well above the perceptual threshold, subjects showed high accuracy (patients 94.75%±2.65; healthy subjects, 96.80%±0.71, mean ± SEM) and sensitivity (d'=1.81 ± 0.27 for patients and 2.12±0.37 for healthy subjects), indicating that the subjects understood and performed the task well following the task rules.

To explore whether visual awareness affects behavioral performance, we divided the trials into 4 groups according to the level of grating contrast and the participants' reported awareness state, that is high-contrast aware (HA), near-threshold-contrast aware (NA), near-threshold-contrast unaware (NU), and low-contrast unaware (LU) (*Figure 2A*, also see Methods). We compared the saccadic parameters between the awareness and unawareness states in near-threshold contrast levels (NA vs. NU), as well as in high and low contrast levels (HA vs. LU), as a control. While saccadic reaction time showed a significant difference between the two awareness states (*Figure 2B*), the other parameters did not. (*Figure 2C*) shows the mean saccadic reaction time of patients and healthy subjects under different awareness states and contrast levels. The results showed that the average saccadic reaction time was significantly shorter under the awareness state than under the unawareness state at both the near-threshold contrast level and the high-low contrast level (p<0.05, Wilcoxon signed rank test), and it was also explicit and consistent at the individual level. The results showed that the saccadic reaction time in the aware trials was systematically shorter than that in the unaware trials. Such results demonstrate that visual awareness significantly affects the speed of information processing in the brain.

### Local field potential results

While the patients were performing our visual awareness task, we recorded the local field potential (LFP) from 901 recording sites (*Figure 2D*, projected on Montreal Neurological Institute (MNI) brain template ICBM152), in which 245 sites were located in the PFC. The table in *Figure 2D* shows a summary of the number of recording sites and patients in different lobes and structures.

### Early visual awareness-related iERP activity in the PFC

While looking through the event-related response of the LFP data, a striking phenomenon in all patients was that the amplitude of LFP showed remarkably different patterns between awareness and unawareness states. We show the grand average (left) and single-trial data (right) of LFP activity of recording sites in the PFC from each patient in *Figure 3A*, which represents the typical awareness-related neural activity of the PFC. There was a vigorous visual evoked response under the HA condi-tion (light red line) compared to the LU (light blue line) condition. However, in the near-threshold

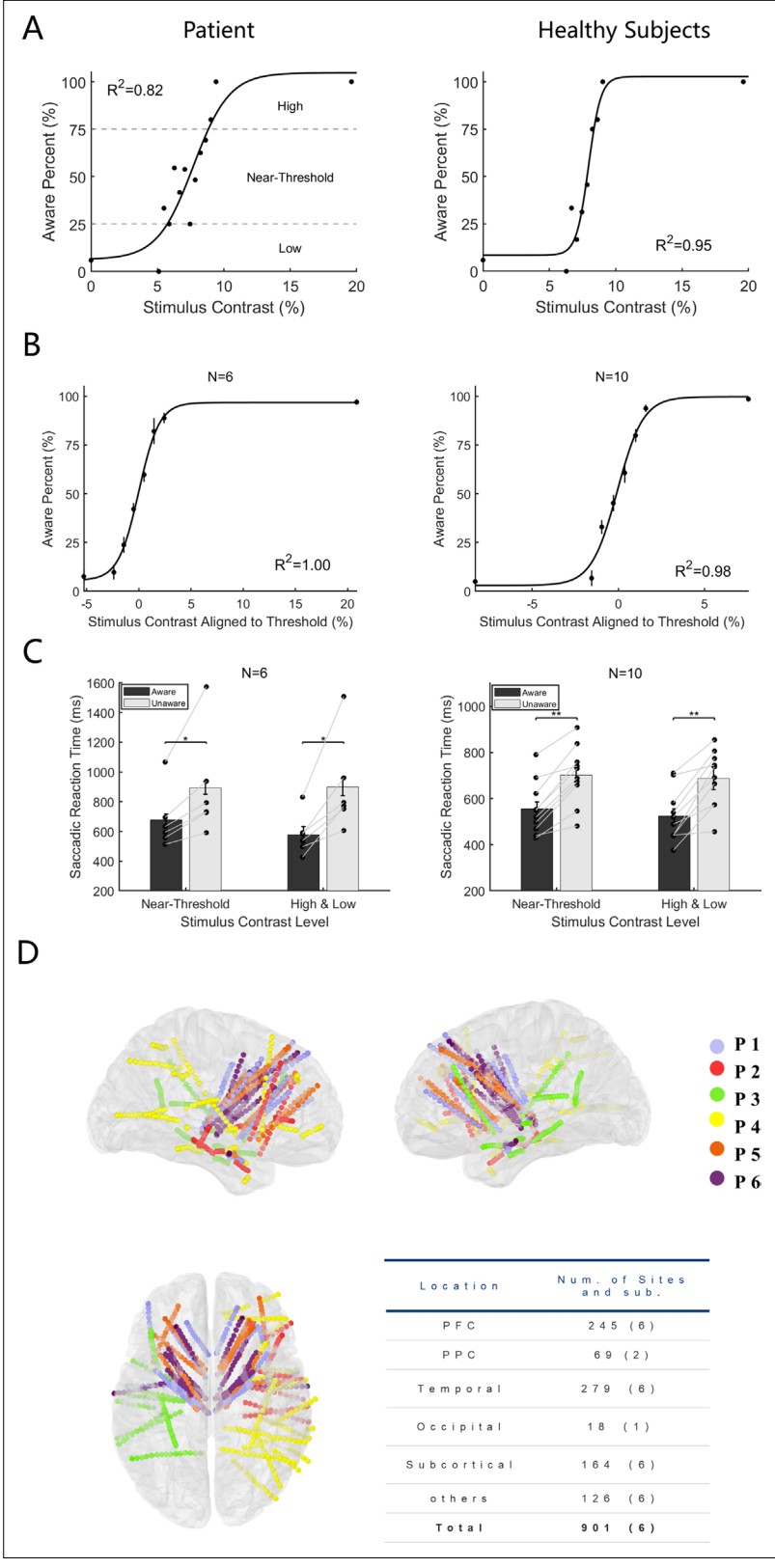

**Figure 2.** Behavioral performance and electrode location. (**A**) Psychometric detection curve in a single session. Left panels from one patient and right from one healthy subject. Each black point in the graph represents the aware percent in a contrast level, and the black curve represents the fitted psychometric function. The contrast level that resulted in an awareness percentage greater than 25%, and less than 75% was defined as

*Figure 2 continued on next page*

*Figure 2 continued*

near-threshold, whereas an awareness percentage less than 25% was low and greater than 75% was high. (**B**) Psychometric detection curves for all participants. The same as Panel A except the contrast is aligned to the individual subject's perceptual threshold, that is, the contrast 0 represents each subject's perceptual threshold. (N, number of subjects; $R^2$, coefficient of determination). (**C**) Saccadic reaction time in the aware and unaware trials under high +low and near-threshold-level contrasts of gratings. Bars show the mean values of saccadic reaction time. In the near-threshold condition, the mean reaction times of patients and healthy subjects in the aware trials were 676.55±42.38 and 556.16±28.45ms, respectively. The mean reaction times in the unaware trials were 892.57±43.28 and 702.30±30.93ms, respectively, and the p values were 0.03 and 9.77x10$^{-4}$, respectively, Wilcoxon signed-rank test. In the high +low condition, the mean reaction times for patients and healthy subjects under the aware trials were 577.78±55.28 and 524.08±31.06ms, and the mean reaction times of the unaware trials were 898.64±57.21 and 687.59±49.07ms, respectively, with p values of 0.002 and 9.77x10$^{-4}$, respectively, Wilcoxon signed-rank test. Each black dot represents one participant. The gray solid line represents the paired dots. Error bars represent the standard error of the mean (SEM). (**D**) Left, right, and top views of all recording sites projected on an MNI brain template. Each color represents a participant. In all brain images, right and up side of the image represent the right and up side of the brain.

The online version of this article includes the following source code for figure 2:

**Source code 1.** Source code files for generating the results in *Figure 2*.

condition, although the contrast of the grating was similar (see Results below), the LFP amplitude of the NA (red line) and NU (blue line) trials started to differ significantly at approximately 200–300ms (thick black line in panels, p<0.01 corrected, by independent t-test, details see Methods). This difference is also very explicit and consistent at the single-trial (>180 trials for NA and NU conditions in individual participants, see Methods) and single-subject level. The divergence onset time (DOT, see Methods) of different recording sites in the PFC is consistently within the period of 200~300ms after grating onset, that is the time window of VAN in scalp EEG mentioned above.

As proposed above, the different activities between NA and NU are regarded as awareness-related activities. We found 89 sites with awareness-related iERP activity in the PFC (323 for all recording sites). *Figure 3B* shows the spatial-temporal dynamics of visual awareness-related activities in all recording sites during the 180–650ms period (sampled at eight time points that equally divided this period, for a clearer display, projected to the cortex, see the *Video 1* for the complete video). Visual awareness-related activities clearly appeared in the PFC during 200–300ms, and these early activities were mainly concentrated in the middle lateral prefrontal cortex (LPFC) and then spread to other brain areas of the PFC. We further calculated the start time of awareness-related activity, that is DOT, of all recording sites. *Figure 3C* shows the spatial distribution of all recording sites DOT (standardized and convenient for visual display), and it can be found that the DOT of the PFC is not explicitly later than those in posterior brain regions, including the posterior parietal lobe, temporal lobe, occipital lobe, etc., and the earliest region in PFC is still located in the middle lateral prefrontal cortex.

Considering the possibility that the brain region with earlier awareness-related activity might be more important for visual awareness, we tried to detect awareness-related activity in the PFC during the 'early' period, that is the VAN time window mentioned above. We focused on the sites in PFC with earlier DOTs. Because of the low-visible stimulus used in this experiment, the onset time of VAN may be delayed to 300–350ms, and we took 350ms as the cutoff value and divided DOT into early (n=43, 48.31%) and late (n=46, 51.69%) phases. We focused on the early DOT sites and calculated the normalized average iERP response of the early DOT sites (*Figure 3D*). At the population level, visual awareness-related activities in the PFC were also significant and started at approximately 200–300ms, which is located in the VAN time window (*Figure 3D*, left). Importantly, there were sites with DOT <350ms from each patient and these sites from different patients were uniformly concentrated in the middle lateral prefrontal cortex (*Figure 3D*, right). That is, the early visual awareness-related activities in the prefrontal cortex are robust and consistent not only at the group level but also at the individual level.

## Early visual awareness-related high-gamma activity in the PFC

Although the above iERP results indicate that there are early visual awareness-related neural activities in the prefrontal cortex, it is difficult to explain whether this response is the result of local processing in the PFC or transmission from other brain regions. Therefore, we calculated the LFP response in the

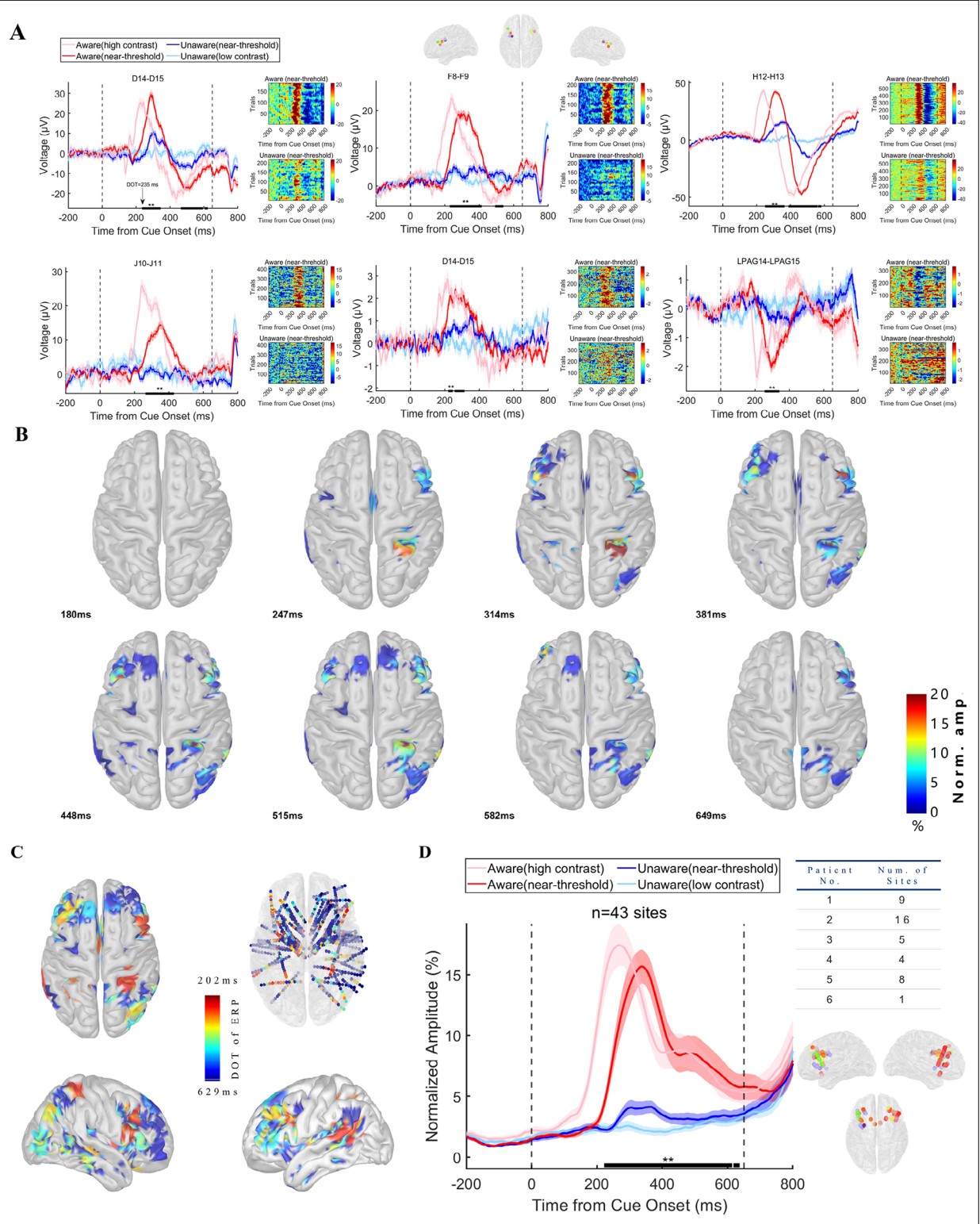

**Figure 3.** Visual awareness-related iERP activities in the PFC. (**A**) LFP activity of example sites from each patient. Each panel represents one example site from one patient. In each panel, the left plot shows the grand average of LFP activity in different conditions. The pink line represents data under HA conditions, the red line represents data under NA conditions, the blue line represents data under NU conditions, and the light blue line represents data under LU conditions. The shaded area of the curve represents the SEM. The two black dotted lines at 0ms and 650ms represent the time when the grating and fixation point change color (the appearance of saccade targets), and the black thick solid line area represents the period when the LFP amplitude is significantly different under NA and NU conditions (p<0.01 corrected, independent sample t-test). The right figure in each panel shows

*Figure 3 continued*

the single-trial data in the NA (upper) and NU (lower) conditions. The color represents the voltage. (**B**) The spatial-temporal distribution of awareness-related ERP activities after the appearance of the grating. Each brain image showed a significant (p<0.01 corrected, see Methods) difference in local field potential in NA and NU trials at all visual awareness-related sites at a specific time point (lower left corner) after the grating appeared. The color represents the standardized voltage difference (see Methods). (**C**) Spatial distribution of divergence onset time. The color represents the normalized DOT value. (**D**) Population results of early phase ERP response in the prefrontal cortex. Different lines represent different conditions, as shown in Panel A. The right table shows the number of recording sites for different patients. The lower right is the location of these sites. The dots with different colors represent different subjects, as shown in *Figure 2D*.

The online version of this article includes the following source code for figure 3:

**Source code 1.** Source code files for pre-processing iEEG data and generating the results in *Figure 3*.

high-gamma (HG) band, which is more representative of the local processing, of the prefrontal cortex. *Figure 4A* shows the exemplified high-gamma activity in the prefrontal cortex from each patient at the grand average (left), spectrogram (middle), and single-trial level (right). It is similar to the iERP results (*Figure 3*) except the data are the HG power and there was a vigorous visual evoked power increase under the HA condition compared to the LU condition. However, in the near-threshold condition, while the contrast of the grating was similar, the high-gamma power of the NA and NU trials started to differ significantly after ~230ms (thick black line in panels, *P*<0.01 corrected by independent t-test; for details, see Methods). This difference is also very explicit and consistent at the single-trial and single-subject levels. The HG activity divergence onset time (DOT) of different recording sites in the PFC was also consistent within the early stage, which was similar to the above iERP result.

We found 31 sites with awareness-related HG activity in the PFC (74 for all recording sites). *Figure 4B* shows the spatial-temporal dynamics of visual awareness-related HG activities in all recording sites during the 0–650ms period (for a clearer display, projected to the cortex, see the *Video 2* for the complete video). Visual awareness-related HG activities also clearly appeared in the PFC during 200–300ms, while they were rarely seen in lower-level visual areas (occipital and temporal lobe), and

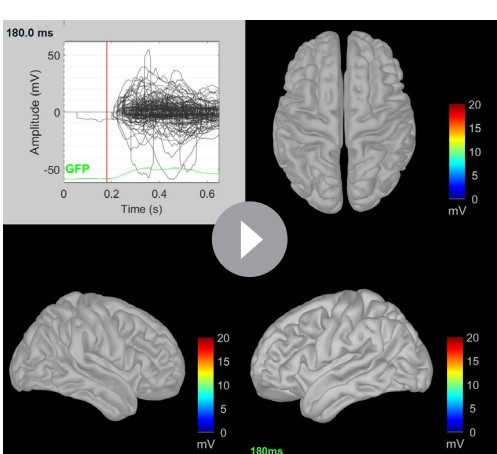

**Video 1.** The spatial-temporal dynamics of visual awareness-related activities (event-related potential (ERP)) in all recording sites during the 180–650ms period (for a clearer display, projected to the cortex). The curves in the upper left panel represent the significant (p<0.01 corrected, see Methods) difference (normalized) in averaged amplitude of local field potential in NA and NU trials at all visual awareness-related ERP sites. The brain image showed the difference projected on the surface at a specific time point after the grating appeared (top, left, and right views were showed at upper right, lower left, and lower right panels, respectively). The color represents the normalized voltage difference (see Methods).

https://elifesciences.org/articles/89076/figures#video1

these early activities were mainly concentrated in the middle lateral prefrontal cortex (LPFC) and then spread to other brain areas of the PFC. We further calculated the start time of awareness-related HG activity, that is DOT, of all recording sites. *Figure 4C* shows the spatial distribution of all recording sites DOT (standardized and convenient for visual display), and it can be found that the DOT of the PFC is not explicitly later than those in posterior brain regions, and the earliest region is still located in the middle lateral prefrontal cortex.

We continue to take 350ms as the cutoff value and divide the HG DOT into early (n=18, 58.06%) and late (13, 41.94%). We focused on the early DOT sites and calculated the normalized mean response (*Figure 4D*) and spectrum of the early DOT sites. At the population level, early visual awareness-related HG activities of the prefrontal sites still exist, and the start time is approximately 200ms, which is within the VAN time window. Importantly, there are sites with HG DOT <350ms for each patient except one, and these sites are also uniformly concentrated in the middle lateral prefrontal cortex. In other words, this effect is significant and consistent at both the group and individual levels. It is worth noting that in the PFC recording sites, the early sites with a high-gamma visual response (7.35%) were explicitly less than

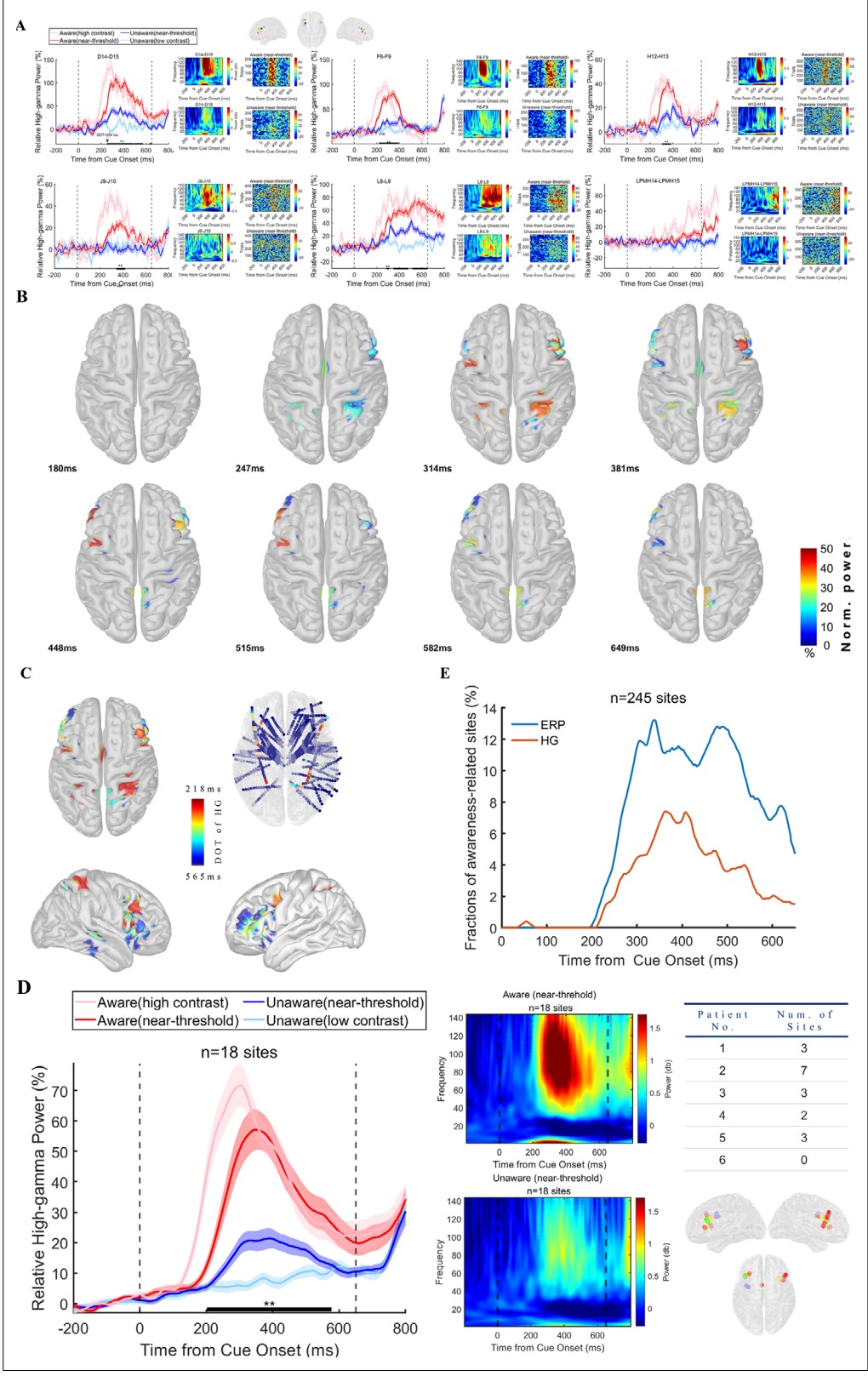

**Figure 4.** Visual awareness-related high-gamma activities in the PFC. (**A**) High-gamma activity of example sites from each patient. Each panel represents one example site from one patient. In each panel, the left figure shows the grand average of high-gamma power in different conditions. The pink line represents data under HA conditions, the red line represents data under NA conditions, the blue line represents data under NU conditions,

*Figure 4 continued*

and the light blue line represents data under LU conditions. The shaded area of the curve represents the SEM. The two black dotted lines at 0ms and 650ms represent the time when the grating and fixation point change color (the appearance of saccade targets), and the black thick solid line area represents the period when the high-gamma power is significantly different under NA and NU conditions (p<0.01 corrected, independent sample t-test). The right figure in each panel shows the single-trial data in the NA (upper) and NU (lower) conditions. The color represents the normalized power. (**B**) The spatial-temporal distribution of awareness-related high-gamma activities after the appearance of the grating. Each brain image showed a significant (p<0.01 corrected, see Methods) difference in high-gamma activity in NA and NU trials at all visual awareness-related sites at a specific time point (lower left corner) after the grating appeared. The color represents the standardized power difference. (**C**) Spatial distribution of divergence onset time for high-gamma activity. The color represents the normalized DOT value. (**D**) Population results of early phase high-gamma response in the prefrontal cortex. Different lines represent different conditions, as shown in Panel A. The middle panel shows the spectrogram in NA (upper) and NU (lower) trials. The right table shows the number of recording sites for different patients. The lower right is the location of these sites. The dots with different colors represent different subjects, as shown in *Figure 2D*. (**E**) Percentage of awareness-related sites in ERP and HG analysis. n, number of recording sites in PFC.

The online version of this article includes the following source code and figure supplement(s) for figure 4:

**Source code 1.** Source code files for generating the results in *Figure 4*.

**Figure supplement 1.** Percentage of awareness-related sites in ERP and HG analysis at parsopercularis and middle frontal gyrus (MFG).

**Figure supplement 1—source code 1.** Source code files for generating the results in *Figure 4—figure supplement 1*.

those with ERP visual response (17.55%), and this is also evident in the temporal dynamics (*Figure 4E*), which may indicate that in the process of visual awareness generation, there may also be a variety of other bands activity for information interaction between multiple brain regions.

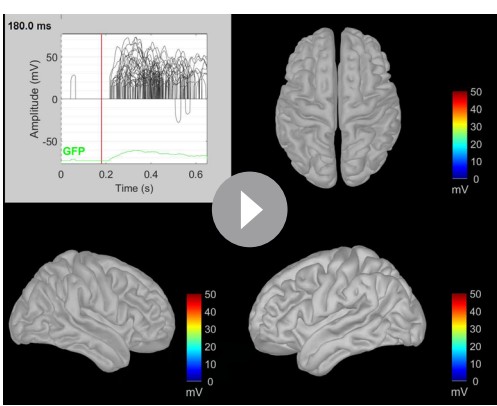

**Video 2.** The spatial-temporal dynamics of visual awareness-related activities (high-gamma (HG) activity) in all recording sites during the 180–650ms period (for a clearer display, projected to the cortex). The same as *Video 1* but for high-gamma activity. The curves in the upper left panel represent the significant (p<0.01 corrected, see Methods) difference (normalized) in averaged magnitude of high-gamma activity in NA and NU trials at all visual awareness-related HG sites. The brain image showed the difference projected on the surface at a specific time point after the grating appeared (top, left and right views were showed at upper right, lower left and lower right panels, respectively). The color represents the normalized magnitude difference (see Methods).

https://elifesciences.org/articles/89076/figures#video2

## Decoding awareness state through ERP and HG activity in the PFC

In addition to the univariate analysis above, we also adopted the machine learning decoding method based on multivariate pattern analysis (MVPA) to test the ability of broadband and high-gamma activities in the PFC to predict the awareness state at the single-trial level. We applied a linear discriminant analysis (LDA) to classify aware versus unaware trials based on broadband and high-gamma LFP activities in the PFC.

*Figure 5A* shows the decoding performance through the ERP activity in the PFC. We found that the ERP activity of the PFC can accurately predict the state of visual awareness at the single-trial level in the delayed period after grating onset, and this result is also relatively consistent at the individual level. Moreover, this decoding accuracy has begun to be much higher than the chance level in 200–300ms, consistent with our above univariate analysis results. Furthermore, we applied the temporal generalization method to test the predictive ability across time. The results (*Figure 5B*) show that the decoding performance of population ERP is very limited in temporal generalization and cannot be well generalized over a long period. Importantly, this phenomenon is also very consistent at the individual level,

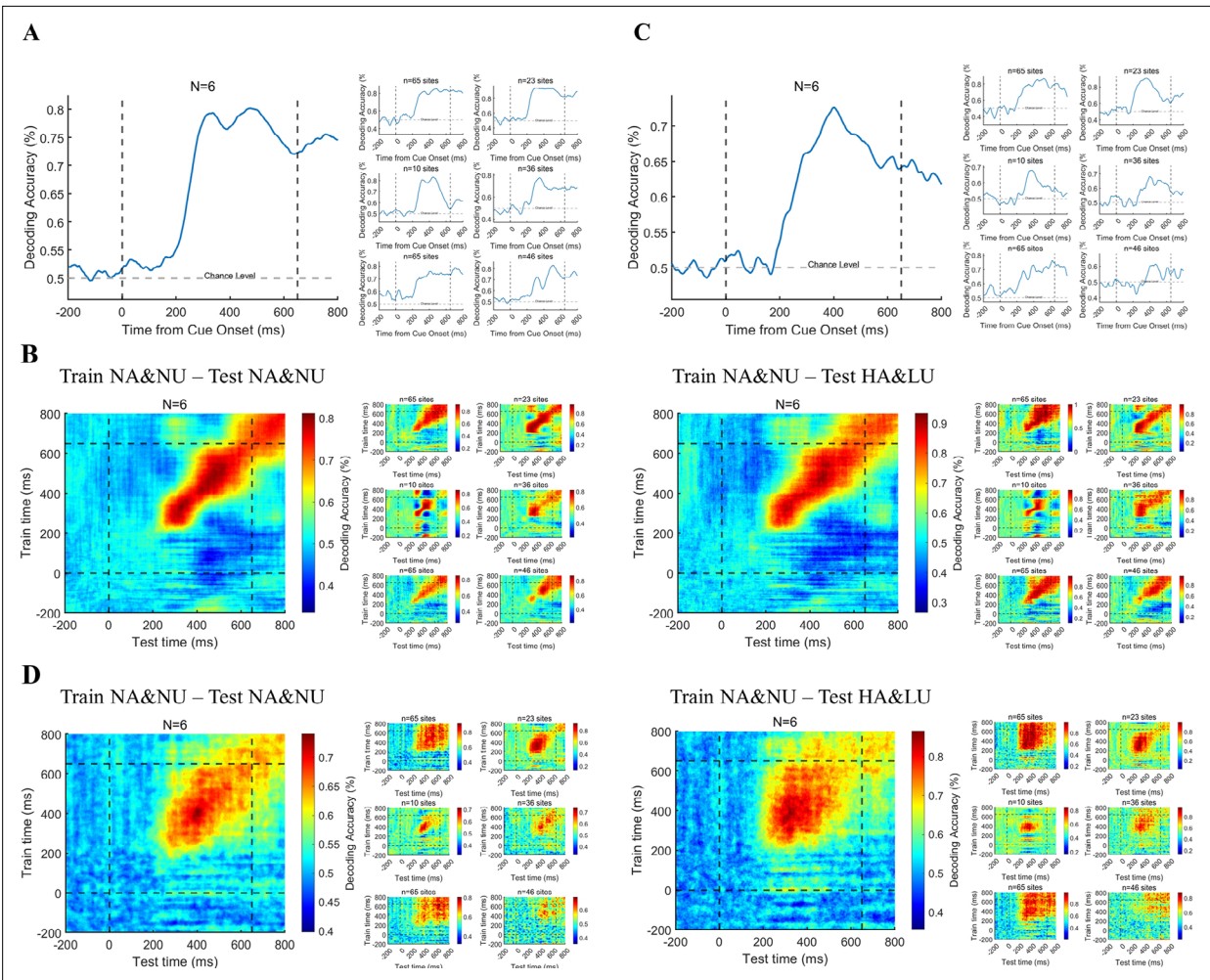

**Figure 5.** Decoding the awareness state through broadband and HG activities. (**A**) Decoding the awareness state through broadband LFP activity. The population result (left) is the average result of six individual subjects (right) after decoding analysis. (**B**) Time generalization of broadband activity decoding. The left panel represents generalization across time. The right panel represents generalization across different conditions, which means training on NA&NU data and testing on HA&LU data. In each panel, the population result (left) is the average result of six subjects (right) after decoding analysis. (**C**) The awareness state is decoded by HG activity. Same as A, except that the HG magnitude is used for decoding. (**D**) Time generalization of HG activity decoding. Same as B, except that the HG magnitude is used for decoding.

The online version of this article includes the following source code for figure 5:

**Source code 1.** Source code files for generating the results in *Figure 5*.

which indicates that the representation of visual awareness via broadband activities in the PFC is relatively dynamic rather than static. Moreover, we also applied temporal generalization across the control conditions, that is train on NA/NU trials but test on HA/LU trials to test the predictive ability across conditions. The classifier trained on the NA/NU trials can be well generalized to the HA/LU trials, and the generalization across time is also very consistent. These results further indicated the robustness of the above decoding results.

*Figure 5C–D* shows the results of a similar analysis of HG activity in the prefrontal cortex. (*Figure 5C*) shows the decoding performance through HG activity in the PFC. We found that the HG activity of the PFC can accurately predict the state of visual awareness at the single-trial level in the delayed period, and this result is also relatively consistent at the individual level. Moreover, this decoding accuracy began to be significantly higher than chance at approximately 200–300ms, which is also consistent with our above univariate analysis results. Furthermore, we applied the temporal generalization method to the HG activity. Although the group results still show that the decoding performance of HG activity is limited in time generalization, the generalization results are not consistent at the individual

level, which may be attributed to the variability of electrode location in different patients. This may suggest that the representation of visual awareness-related information by population neurons at different regions in the PFC is inconsistent, that is some are relatively stable, and others are dynamic. Similarly, we applied temporal generalization across the HA/LU conditions, and the result was also very consistent.

## Dynamic functional connectivity changes in the PFC associated with visual awareness

We further explored the dynamic functional connectivity between the prefrontal cortex and other brain regions in the generation of visual awareness. First, we calculated the time-across phase-locking value (PLV) (*Tass et al., 1998*) in NA and NU conditions at the sensor level (baseline removed, see Methods for details). *Figure 6A–B* shows the example results of two exemplified patients (see *Figure 6—figure supplement 1*. for other patients). We found that in the low-frequency band (1–8 Hz), overall, there was a significant difference in the PLV between the NA and NU trials. In the first 150ms after the appearance of the grating, there was no significant difference in the PLV between these two conditions. However, in the period of approximately 200–300ms, the PLV between many recoding sites in the NA trials began to increase explicitly and lasted until after 600ms. In contrast, there was no such phenomenon in the NU trials. Importantly, this phenomenon is strikingly explicit and consistent in each patient.

Furthermore, we grouped the recording sites by brain regions according to the DKT brain atlas *Fischl, 2012* and averaged the PLV between different sites and different subjects according to brain regions (*Figure 6C*). The functional connectivity between the prefrontal cortex and subcortical structures, such as the thalamus, and the posterior cortex, such as the posterior parietal cortex, was significantly enhanced in the NA trials compared with the PLV in NU trials. In addition, the functional connectivity between the brain regions within the PFC was also significantly enhanced. For clearer visualization, we show the dynamics of PLV in the NA and NU trials on a topographic map (*Figure 6D*).

Interestingly, we found that at the single-subject level, the sites in the prefrontal cortex that showed early awareness-related PLV activity coincided with the above sites with early awareness-related HG activity. *Figure 6E* shows the averaged PLV between these sites (n=18) with early visual awareness-related HG activity in the PFC and other sites in the NA and NU trials. The averaged PLV also showed significant visual awareness-related activities. Moreover, at the population level, the start time of awareness-related PLV activity at these sites also started at approximately 200ms, which is very similar to that of HG activity.

## The different neural activity between NA and NU trials is not caused by the grating contrast

Since there may be a slight difference in the grating contrast under the NA and NU trials, the aforementioned different activity between the NA and NU conditions may be caused by the different contrast of grating. To rule out this possibility, we compared the distribution of grating contrast in the NA and NU conditions for each patient (*Figure 7A*). Overall, we find that the distributions of grating contrasts largely overlap with each other (red and blue curves). The Kolmogorov–Smirnov test resulted in no significant difference (p>0.05) between the two contrast distributions in all patients except patient 2. Although intuitively the trials with higher contrast contributed more to the NA condition and the trials with lower contrast contributed more to the NU condition, the contrast difference between the NA and NU conditions were not significantly different perhaps because the grating contrast was designed to be near the perceptual threshold (the 'aware' percent is 50%) in most trials. Moreover, for patient 2, the difference between the mean grating contrast in the NA and NU trials was also subtle (0.15%), much less than the difference between NA and HA (17.35%) or NU and LU (0.84%). However, it is clear in the results above that the different activities between NA and NU were much stronger than those between the same awareness states, that is NA vs. HA or NU vs. LU. Therefore, the difference in neural activity between aware and unaware trials cannot be explained by the contrast differences in the grating.

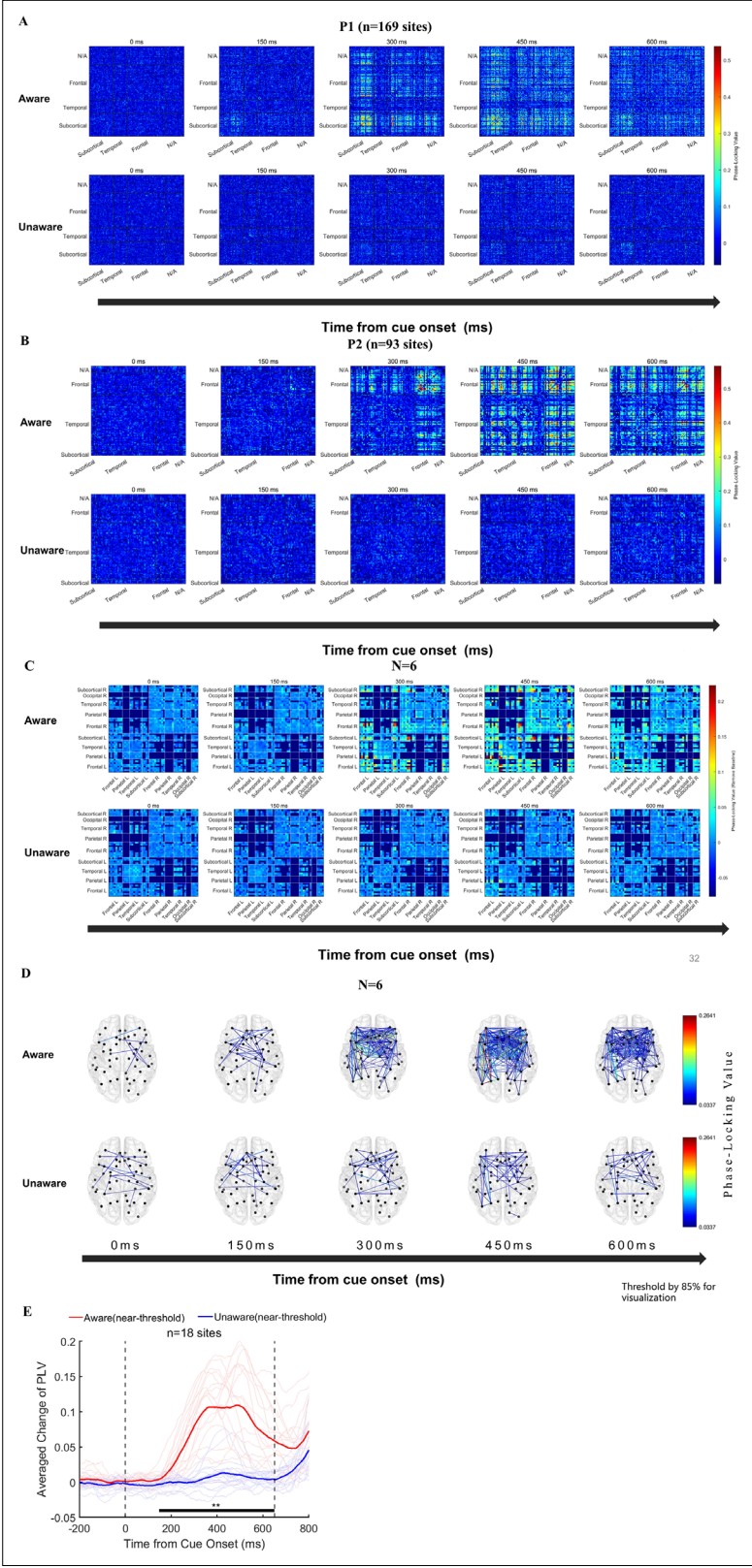

**Figure 6.** Functional connectivity analysis results. (**A–B**) Phase-locking value (PLV) changes at the sensor level (NxN) in two example patients. From left to right, the changes in PLV between each recording site at the five time points of 0/150/300/450/600ms after the appearance of the grating in the near-threshold aware (upper) and near-threshold unaware (lower) trials are displayed (baseline removed, see the Methods). The color represents

*Figure 6 continued on next page*

*Figure 6 continued*

PLV. (**C**) The population results of functional connectivity analysis averaged according to brain regions. From left to right, PLV between different brain regions at the five time points of 0/150/300/450/600ms after the appearance of the grating in the near-threshold aware (upper) and near-threshold unaware (lower) trials (baseline removed, see the Methods). The color represents the PLV. (**D**) For the same data as C, only the strongest 15% PLV is displayed on the brain template for clear visualization. (**E**) PLV under different conditions at early-phase HG awareness-related sites in the PFC. The red line represents the average PLV in the NA trials, and the blue line represents the average PLV in the NU trials. The faded line represents the PLV of each site.

The online version of this article includes the following source code and figure supplement(s) for figure 6:

**Source code 1.** Source code files for generating the results in *Figure 6* and *Figure 6—figure supplement 1*.

**Figure supplement 1.** Functional connectivity analysis results of Patient 3–6.

## Discussion

We employed a novel visual awareness task and found that, first, the saccadic reaction time of all subjects was shorter when they were aware of grating than when they were not aware. Second, there are ERP and HG activities related to visual awareness in the prefrontal LFP, and these activities begin at the early stage (200–300ms). Then, we found that the awareness state can be reliably decoded from the broadband and HG activities of the PFC at approximately 200–300ms after stimulus onset. Finally, the functional connectivity between the PFC and other brain regions in the low-frequency band (1–8 Hz) also appeared awareness-related changes, and the initial time and location of this change are consistent with the above HG results. The results above are not caused by the difference in the physical properties of external stimuli. Therefore, we propose that the prefrontal cortex also plays a key role in the early stage of visual awareness generation even if motor effects are excluded.

### Advantages and limitations of the behavioral paradigm used in the present study

Compared with the regular report-dependent visual awareness task, we employed a task that effectively dissociated the activity of visual awareness from the activity of saccadic decision and execution in terms of time (the fixation point changes color 650ms after the grating onset) and reporting behavior. Since the choice of reporting behavior depends on the visual awareness state and the color

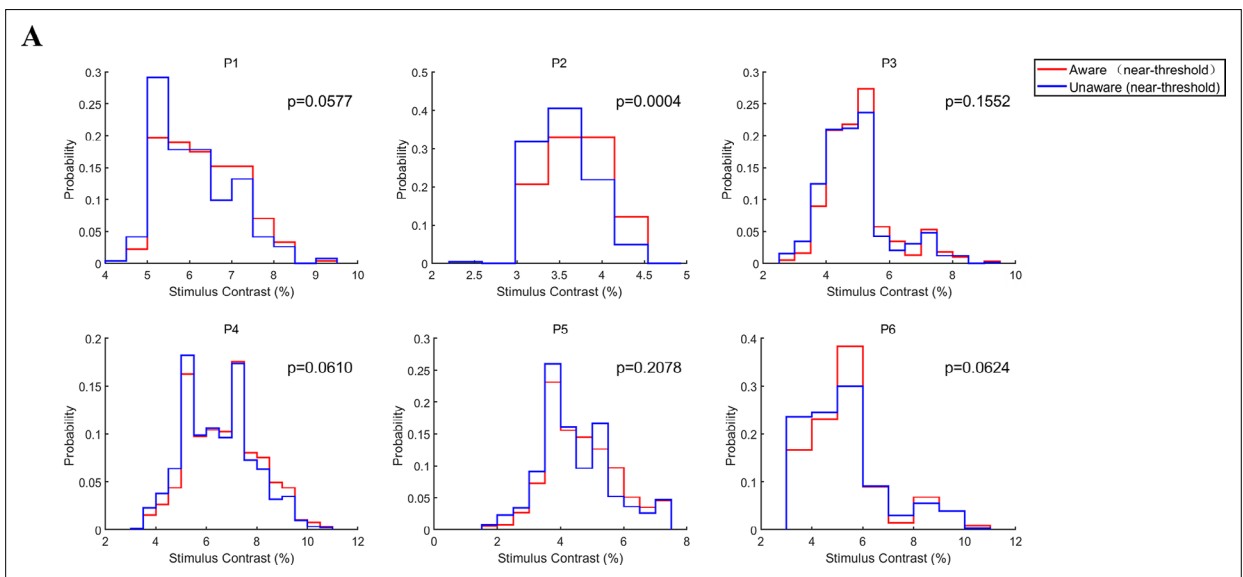

**Figure 7.** Grating contrast in NA and NU conditions. (**A**) Distribution of grating contrast in the NA and NU conditions for each patient. Each panel represents one patient. The red and blue lines represent the distribution of grating contrast in NA and NU conditions, respectively.

The online version of this article includes the following source code for figure 7:

**Source code 1.** Source code files for generating the results in *Figure 7*.

of the fixation point, the subjects were unable to effectively prepare the subsequent saccadic reports during the delay period of the task; thus, the visual awareness-related activity evoked by the grating stimulus and report-preparation-related activity were temporally dissociated. Furthermore, both in the awareness and unawareness states, the subjects needed to choose the saccade direction after the fixation point changes its color so that the saccade directions in the two visual awareness states were fully matched to exclude the effects of saccade preparation- and execution-related activity on visual awareness-related activity. Compared with the no-report paradigms, our novel paradigm required subjects to make explicit reports of their awareness states so that it can be more accurate regarding the judgment of the awareness status.

Moreover, the intervals for the initial fixation period and delay period were both fixed at 600ms, instead of randomly varied, in order to keep subjects' anticipation being similar in all trials. Also, the location of grating remained same in all trials for the same purpose.

It was also worth noting that our task does not remove the confound of report entirely, that is the post-perceptual confound might relate to planning to report perception, which is different for perceived and not perceived stimuli. Nevertheless, our task did minimize the motor-related confound to awareness-related activity.

## Visual awareness states affect saccadic reaction time

Many factors affect saccadic reaction time, including the characteristics of external stimuli and the internal state of the brain. The physical properties of external visual stimuli have a direct impact on the performance of saccades; for example, saccadic reaction time decreases with increasing visual stimulus contrast (*Allik and Kreegipuu, 1998*; *Deplancke et al., 2010*). A widely accepted view is that the greater contrast of visual stimuli will automatically attract more bottom-up attention and speed up information processing in the brain, thereby shortening the saccadic reaction time (*Posner and Petersen, 1990*; *Corbetta and Shulman, 2002*). Other studies found that endogenous factors, such as top-down attention, can also significantly shorten the saccadic reaction time (*Corbetta and Shulman, 2002*; *Posner, 1980*).

One study briefly reported that the average reaction time (button press) for word semantic classification in the awareness state was approximately 300ms faster than in the unawareness state, but the statistical test did not reach a significant level (*Gaillard et al., 2009*). In present study, we found that the saccadic reaction time of all subjects in the aware trials was significantly smaller than that in the unaware trials (*Figure 2C*). More critically, the grating contrast near the threshold was not significantly different between the aware and unaware trials (*Figure 7A*); thus, such results cannot be merely caused by the contrast difference in the grating. Here, we showed that awareness states significantly affect saccadic reaction time, which supports our previous hypothesis.

An alternative interpretation for RT difference between aware and unaware condition in our study is that the difference in task-strategies used by subjects/patients to remember the response mapping rules between the perception and the color cue (e.g. if the YES +GREEN = RIGHT and YES +RED = LEFT rules were held in memory, while the NO mappings were inferred secondarily rather than being actively held in memory).

Another possibility is that the reaction time is strongly modulated by the confident level, which has been described in previous studies (*Marzi et al., 2006*; *Broggin et al., 2012*). However, in previous studies, the confident levels were usually induced by presenting stimulus with different physical property, such as spatial frequency, eccentricity and contrast. However, the dependence of visual process on the salience of visual stimulus confounds with the effect of visual awareness on the reaction time of responsive movements, which is hard to attribute the shorter reaction time in more salient condition purely to visual awareness. In contrast, we create a condition (near aware threshold) in the present study, in which the saliency (contrast) of visual stimulus is very similar in both aware and unaware conditions in order to eliminate the influence of stimulus saliency in reaction time. We think that the difference in reaction time in our study is mainly due to the modulation of awareness state, which was not reported previously.

## Visual awareness-related activities appeared in the PFC at an early stage

The long debate about the neural mechanism of consciousness focuses on the location, that is 'front' vs. 'back', and the time, that is 'early' vs. 'late', of its origin in the brain (*Seth and Bayne, 2022*). Concerning the dispute of location, our results show that the prefrontal cortex still displays visual awareness-related activities even after minimizing the influence of the motor-related confounding variables related to subjective reports such as motion preparation, which indicates that the prefrontal cortex does participate in the information processing of visual awareness. Regarding the time dispute, our results show that at 200–300ms after stimulus onset, the VAN time window, the prefrontal cortex has activities related to visual awareness, and these activities include not only ERP activities but also HG activities, which are highly related to the firing of local population neurons (*Fisch et al., 2009*). Notably, the recording sites with early ERP and HG visual awareness-related activities are mostly concentrated in the middle lateral prefrontal cortex. This shows that the prefrontal cortex, especially the middle lateral prefrontal cortex, started visual awareness-related activities, including local activities, at a relatively early stage.

Several iEEG studies have shown the early prefrontal cortical involvement in visual perception, including ERP and HFA (*Blanke et al., 1999*; *Khalaf et al., 2023*; *Vishne et al., 2023*). However, in these studies, the differential activity between conscious and unconscious conditions was not investigated, thus, the activity in prefrontal cortex might be correlated with unconscious processing, rather than conscious processing. In present study, we compared the neural activity in PFC between conscious and unconscious trials, and found the correlation between PFC activity and conscious perception.

Although one iEEG study (*Gaillard et al., 2009*) reported awareness-specific PFC activation, the awareness-related activity started 300ms after the onset of visual stimuli, which was ~100ms later than the early awareness related activity in our study. Also, due to the limited number of electrodes in the previous study (2 patients with 19 recording sites mostly in mesiofrontal and peri-insular regions), it was restricted while exploring the awareness-related activity in PFC. In the present study, the number of recording sites (245) were much more than previous study and covered multiple areas in PFC. Our results further show earlier awareness-related activity (~200ms after visual stimuli onset), including ERP, HFA and PLV. These awareness-related activity in PFC occurred even earlier (~150ms after stimulus onset) for the salient stimulus trials (*Figure 3A, D* and *Figure 4A, D*, HA condition)'.

However, the proportions are much smaller than that reported by Gaillard et al, which peaked at ~60%. We think that one possibility for the difference may be due to the more sampled PFC subregions in present study and the uneven distribution of awareness-related activity in PFC. Meanwhile, we noticed that the peri-insula regions and middle frontal gyrus (MFG), which were similar with the regions reported by Gaillard et al, seemed to show more fraction of awareness-related sites than other subregions during the delay period (0–650ms after stimulus onset). To test such possibility and make comparison with the study of Gaillard et al., we calculated the proportion of awareness-related site in peri-insula and MFG regions (*Figure 4—figure supplement 1*). We found although the proportion of awareness-related site was larger in peri-insula and MFG than in other subregions, it was much lower than the report of Gaillard et al. One alternative possibility for the difference between these two studies might be due to the more complex task in Gaillard et al. Nevertheless, we think these new results would contribute to our understanding of the neural mechanism underlying conscious perception, especially for the role of PFC.

Some recent studies utilizing the no-report paradigm supported our results. For example, a recent electrophysiological study of rhesus monkeys found that even if monkeys do not need to report their conscious content under the binocular competition paradigm, there were also visual awareness-related spiking activities in the prefrontal cortex (*Kapoor et al., 2022*). In addition, some previous human intracranial studies also reported ERP, HG, and single-neuron activities related to visual perception in the higher visual areas of the temporal lobe, such as the superior temporal gyrus (*Fisch et al., 2009*; *Reber et al., 2017*). However, due to the limitations of the experimental paradigm, these studies cannot completely exclude the impact of subjective reports. Moreover, most of the electrodes in previous studies were located in the temporal lobe, and electrodes in the prefrontal cortex are rare. As our data show that the proportion of sites with earlier ERP and HG responses in the prefrontal cortex is low and relatively concentrated, it is not hard to explain that earlier visual awareness-related activities in the prefrontal cortex were not found in previous iEEG studies, especially in the HG band.

Our results, on the one hand, support that the prefrontal cortex may play an important role in the emergence of visual awareness and thus support the 'front of the brain' theories of consciousness in the origin brain regions. However, our data also support the prediction of the early emergence of consciousness proposed by the 'back of the brain' theories. This means that our result also challenges some predictions of these mainstream theories.

## Awareness state can be effectively decoded from PFC activities from an early stage

Compared with traditional univariate analysis, MVPA-based machine learning decoding may be more sensitive to the information contained in brain activities (*Grootswagers et al., 2017*), so it has gradually become a powerful tool for the study of neural correlates of consciousness (NCCs). Therefore, we used linear discriminant analysis (LDA) to decode the awareness states at the single-trial level through broadband and HG activities in the prefrontal cortex. Our decoding results show that whether through broadband or HG activities, the decoding performance began well above the chance level at approximately 200–300ms. This shows that the prefrontal cortex has begun to encode visual awareness-related information in 200–300ms, which is consistent with the above univariate analysis results. In addition, a recent electrophysiological study of rhesus monkeys using the no-report paradigm also found that even without the requirement of explicit reporting, awareness-related neurons can still be found in the lateral prefrontal cortex of rhesus monkeys, and conscious content can be accurately decoded by the MVPA method (*Kapoor et al., 2022*). Another human fMRI study also found similar results (*Kronemer et al., 2022*). These results supported our experimental results.

In addition, we also applied the temporal generalization method during decoding. Temporal generalization analysis shows how different brain activity patterns change and transform over time (*King and Dehaene, 2014*). Our results show that the representation pattern of neural activity in the prefrontal cortex for visual awareness-related information is dynamic, not static, and this result is very consistent at the individual level and across the control conditions. Such a result may be consistent with the nature of conscious perception. The process of conscious perception (visual awareness) includes a series of processes from phenomenal consciousness ('qulia') to conscious access. Therefore, at the level of large, high-order brain regions, the representation of this complex process in the brain should be continuously transmitted and transformed, that is, dynamically changed.

## Evaluate the role of the prefrontal cortex in visual awareness at the brain network level

There is growing evidence showing that conscious experience may be represented at the brain network level (*Kronemer et al., 2022*; *Huang et al., 2023*). Thanks to the simultaneously recorded LFP data in multiple brain regions with a high signal-to-noise ratio and high spatial-temporal resolution, we can analyze the dynamic functional connectivity changes under the aware and unaware conditions at the millisecond scale. Since the signals in the low-frequency band are more involved in the information interaction between different brain regions and the activities in the high-frequency band are more representative of the local neural activities, it is easy to understand the functional connectivity changes in the low-frequency band between relatively distant brain regions. Some previous studies have found that the phase synchronization of neural activities in beta and gamma bands is related to visual awareness, but because most of these studies are based on fMRI, it is difficult to describe the dynamic changes in such brain networks on a more precise time scale. The only human iEEG study (*Gaillard et al., 2009*) reported that the phase synchronization of the beta band in the aware condition also occurred relatively late (>300ms) and mainly confined to posterior zones but not PFC. Moreover, most of these studies did not rule out the influence of motor-related processing, and these processes are often associated with changes in the functional connectivity of multiple brain regions. To the best of our knowledge, we are the first to report dynamic changes in functional connectivity on the millisecond time scale in the low-frequency band (1–8 Hz) associated with visual awareness, and this change is not caused by motor-related confounding factors, such as motion preparation. Moreover, the early sites of PLV and HG overlapped each other to a large extent. Such converging results may further suggest that the middle lateral prefrontal cortex, the brain regions where these sites are located, plays a crucial role in the emergence of visual awareness.

## Our results may contribute to some theoretical concepts related to consciousness

Compared with the unaware trials, the high-frequency activity in the PFC and the functional connectivity between many brain regions in the aware trials increased. These results may confirm some theoretical concepts, such as 'ignition' and conscious access. Global neuronal workspace theory (GNWT) emphasizes ignition and conscious access (*Dehaene and Changeux, 2011*; *Mashour et al., 2020*). Ignition means nonlinear brain activation to small stimulus changes (such as perceptual threshold stimulation). For the near-threshold stimulus used in the current study, while the sensory information in the brain passes through a certain threshold in a specific brain area, such as the lateral prefrontal cortex, the population of neurons discharges (HG activity of local field potential), that is, 'ignition'. This is consistent with many previous human intracranial studies (*Fisch et al., 2009*; *Reber et al., 2017*). However, as discussed above, in contrast with previous studies, our study detected earlier awareness-specific 'ignition' in the human PFC, while minimizing the motor-related confounding.

Conscious access means that conscious contents can be shared by many other local information processors in the brain (*Dehaene and Changeux, 2011*; *Mashour et al., 2020*). For example, when you see the grating, you can either report it orally or draw it down. Such a function may require the participation and coordination of many related brain regions or networks, but there is no clear biomarker for how these brain regions coordinate. The explicit increase in the PLV, specifically appearing in the aware trials, in the low-frequency band found in the current study may provide potential biological meaning for conscious access; that is, many brain regions interact with each other through phase synchronization in the low-frequency band to accomplish information sharing of conscious content.

Interestingly, the beginning time of prefrontal visual awareness-related activities found in ERP and HG activities is very consistent with the onset time of dynamic functional connectivity enhancement, both of which start at 200–300ms. In addition, the dynamic functional connectivity of the sites with early HG visual awareness-related activity is usually the earliest to enhance compared to other sites, and the two have good consistency. And this consistency suggested that conscious access and phenomenal awareness may be closely coupled, occurring initially in the lateral PFC at the early stage, and indicate a potential process of conscious perception: subtle change in the external stimulus or internal brain state - population neuron discharge in specific brain areas (ignition, phenomenal consciousness appeared) - phase synchronization of various brain areas induced by specific brain areas (conscious access). Further research is needed to regulate the activity of the middle lateral prefrontal cortex in the high-gamma band or the phase synchronization in the low-frequency band to explore whether the prefrontal cortex plays a causal role in the process of conscious perception.

## Methods

### Data acquisition

Six adult patients with drug-resistant epilepsy (6 males, 32.33±4.75 years old, mean ± SEM) and 10 healthy subjects (4 males, 6 females, age 27.80±3.92, mean ± SEM) participated in this study. All subjects had normal or corrected-to-normal vision. Electrophysiological signals (LFPs) were obtained from six patients. Stereotaxic EEG depth electrodes (SINOVATION MEDICAL TECHNOLOGY CO., LTD. Ltd., Beijing, China) containing 8–20 sites were implanted in patients in the Department of Neurosurgery, PLA General Hospital. Each site was 0.8 mm in diameter and 2 mm in length, with 1.5 mm spacing between adjacent sites. A few electrodes were segmented, and the distance between the two segments was 10 mm. Electrode placement was based on clinical requirements only. Recordings were referenced to a site in white matter. The sEEG signal was sampled at a rate of 1 kHz, filtered between 1 and 250 Hz, and notched at 50 Hz (NEURACLE Technology Co., Ltd., Beijing, China). Stimulus-triggered electrical pulses were recorded simultaneously with sEEG signals to precisely synchronize the stimulus with electrophysiological signals.

Ten healthy subjects performed the experiment in a quiet dark room. Stimuli were presented on a 27-inch LED screen (BenQ, refresh rate of 120 Hz, resolution of 1920 x 1080), and eye movement data were recorded by an infrared eye tracker (EyeLink 1000 Desktop; SR Research) with a sampling rate of 1 kHz. Patients performed the experiment in a quiet dark environment. The stimuli were presented on a 24-inch LED screen (Admiral Overseas Corporation, refresh rate of 144 Hz, resolution of 1920 x 1080), and eye position data were obtained by an infrared eye tracker (Jsmz EM2000C, Beijing Jasmine

Science & Technology, Co. Ltd), sampling at 1 kHz. The experimental paradigm was presented by MATLAB (The MathWorks) and the toolbox Psychtoolbox-3 (PTB-3; Brainard and Pelli). Patients were regular recipients of standard medication for epilepsy treatment during the experimental period.

All subjects provided informed consent to participate in this study. The Ethics Committee of Chinese PLA General Hospital approved the experimental procedures (approval numbers S2022-457-01).

## Electrode localization

Electrode locations were determined by registering postoperative computed tomography (CT) scans and preoperative T1 MRI using a standardized mutual information method in SPM12 (*Ashburner and Friston, 1997*). Cortical reconstruction and parcellation (Desikan-Killiany Atlas) were conducted by FreeSurfer (*Fischl, 2012*). To demonstrate the MNI coordinates of recording sites, a nonlinear surface registration was performed using a nonlinear registration in SPM12, aligning to the MNI ICBM152 template.

## Experimental task

A trial started when a fixation point (0.5° x 0.5°, white cross) appeared at the center of the screen (radius of eye position check window is 4°, the dotted circle). After the subject fixated on the fixation point for 600ms, a cue stimulus (Gabor grating, 2x2° circle) was presented for 50ms at a fixed position (7°) on the left (or right) side of the screen for all participants. The side where the grating appears (left or right) is opposite to the hemisphere where the patients' electrodes are implanted (left or right). For example, if the patient's electrodes are implanted on the right side, the grating would appear on the left side of the screen. For patients 1, 5, and 6, the electrodes were implanted in both hemispheres, and the grating is set on the right side. In 70% of the trials, the grating contrast (weber contrast, see *van Vugt et al., 2018*) was maintained near the subject's perceptual threshold by a 1 up/1 down staircase method, and the step was either 0.39% or 0.78% for each participant. In 10% of the trials, the stimulus contrast was well above the threshold, and in the other 20% of the trials, the stimulus contrast was 0, namely, no stimulus appeared. After another 600ms delay, the color of the fixation point turned red or green, and two saccade targets (1° x 1°, white square) appeared at fixed positions (10°) on the left and right sides of the screen. If the grating was seen, the green fixation point was required to make a saccade to the right target, while the red fixation point was required to make a saccade to the left target. If the grating was not seen, the rule of saccadic direction was inverted. Gratings with high contrast (well above the perceptual threshold) and zero contrast (grating absent) served as control conditions to evaluate the understanding and performance of the task. Before starting to collect data, subjects accepted 1–2 training sessions, and the contrast perceptual threshold was determined for each subject. In the formal experiments, each session consisted of 180 trials, and the intertrial interval (ITI) was 800ms.

### Data analysis

Each healthy subject completed two sessions, and each patient completed five to seven sessions. Each session contained 180 trials in total. We excluded trials in which subjects broke fixation during the fixation period (eye position exceeded 4x4° check window by more than 40ms) or the saccadic latency exceeded max response duration (2000ms for most patients, 5000ms for patient 6 according to the patient's performance) after the target appeared. For all subjects, more than 85% of the trials were available for further analysis.

In each session, according to the subjects' reports of being aware and unaware of the existence of grating, we divided the trials into awareness and unawareness states. According to the percentage of subjects reporting 'awareness', we further divided grating contrast into three levels: high contrast (aware percentage >75%), near-threshold contrast (25% ≤ aware percentage ≤ 75%), and low contrast (aware ratio <25%). Due to the few trials of low-contrast-aware and high-contrast-unaware trials, we next analyzed only trials under four conditions of awareness: low-contrast-unaware (LU), near-threshold-unaware (NU), near-threshold-aware (NA) and high contrast-awareness (HA). In total, under LU/NU/NA/HA conditions, there were 243/353/343/160 trials (22.1/32.1/31.2/15.6%, averaged across patients). Since we were more concerned with the differences in subjects' behavior and electrophysiological responses during the physical stimulus, we focused on the results under NA and NU conditions

in the subsequent analysis. The results under LU and HA conditions were classified as the control group and were only used to verify and check the results during calculation.

## Behavioral data analysis

Psychophysical curve fitting uses the following formula:

$$AP = a + \frac{b}{1 + exp\left(-d * \left(SC - c\right)\right)}$$

AP, ratio of reported awareness; SC, stimulus contrast; a-d, individual fitting parameters.

For population psychometric curve fitting, adjacent contrast levels were averaged, resulting in 8 contrast levels in each session.

## Saccadic latency calculations

The saccadic latency was calculated according to a custom program. Eye position and saccade velocity were smoothed with 20ms sliding windows, and each step was 1ms. Saccade onset time was calculated as the time when the saccade velocity first exceeded 30°/s and lasted more than 20ms. Saccadic latency was defined as the time interval between the onset of the saccadic target and the onset of the saccade.

## LFP data analysis

### Data preprocessing

Data preprocessing was performed by the software package EEGLAB (*Delorme and Makeig, 2004*) in MATLAB. Data were filtered between 1–250 Hz and notch-filtered at 50, 100, 150, 200, and 250 Hz with an FIR filter. Epochs were extracted from –490ms before grating onset to 1299ms after grating onset. Bad recording sites were discarded for analysis based on visual inspection and power spectral density (PSD) analysis. each recording site was applied a bipolar reference, rereferenced to its direct neighbor.

### Spectral analyses

Time-frequency decomposition in high-gamma bands (60–150 Hz) was achieved by Morlet wavelets in the Brainstorm toolbox (*Tadel et al., 2011*). Before computation, the mean ERP in each condition was removed. For visualization, we used a prestimulus baseline correction over –200–0ms, and the results in *Figure 4* show the HG activity increases or decreases relative to this baseline. To avoid windowing artifacts, we only reported results of –200–800ms.

## Quantitative definition of visual awareness-related activity in recording sites

We first identified visual awareness-related sites. For all recording sites, we compared the broadband amplitude (for ERP analysis) or high-gamma magnitude (for HG analysis) of LFP between the NA and NU conditions within 650ms after grating onset by statistical tests for each time point. We defined the site where there was a significant difference (p<0.01 FDR corrected for time points and channels, independent t-test) in the LFP between NA and NU conditions and persisted for more than 20ms as an awareness-related site. We also defined the start time of divergence as the divergence onset time (DOT) and defined the mean of the LFP amplitude difference in the first divergence period as the divergence amplitude (DA).

Normalized DOT:

$$T_d = T_{target} - DOT$$

$$normDOT = \frac{T_d - T_{dmin}}{T_{dmax} - T_{dmin}}$$

DOT, divergence onset time; $T_{target}$, time from target onset to grating onset, 650ms; $T_d$, time from DOT to $T_{target}$; $T_{dmin}$, the minimum value of $T_d$ in all awareness-related sites within one patient; $T_{dmax}$, the

maximum value of $T_d$ of all awareness-related sites within one patient; normDOT, normalized DOT value, a larger normDOT value means a shorter DOT.

For population analysis of recording sites, we conducted baseline normalization for each site.

Baseline normalization for ERP amplitude:

$$x_{norm} = \frac{x - \mu}{\mu} * 100$$

$x_{norm}$, normalized value; x, LFP amplitude; μ, mean LFP amplitude in the baseline period (−200–0ms).

Since it is not clearly understood so far regarding the biological characteristics of LFP polarity (*Einevoll et al., 2013*), to simplify such complex issue, we consider the change in magnitude of LFP during delay period in our task is awareness related activity, regardless its actual value being positive or negative. Therefore, while analyzing the awareness related population activity, we first calculate the absolute value of activity difference between aware and unaware trials in individual recording site, then pool the data of 43 recording sites together and calculate the mean and standard error of mean (SEM) (*Figure 3D*). As in *Figure 3A*, the activity difference between aware (red) and unaware (blue) trials lasts until/after the end of delay period. Thus, the awareness related population activity in *Figure 3D* extends out to 600ms.

Baseline normalization for HG magnitude:

$$x_{norm} = \frac{x - \mu}{\sigma} * 100$$

$x_{norm}$, normalized value; x, high-gamma magnitude or power; μ, mean of high-gamma magnitude or power in the baseline period (−200–0ms); σ, standard error of high-gamma magnitude or power in the baseline period (−200–0ms).

Baseline normalization for spectrogram:

$$x_{norm} = 10 * log10\left(x/\mu\right)$$

$x_{norm}$, normalized value; x, power value; μ, mean power in the baseline period (−200–0ms).

## Topographic display of ERP and HG data

To visualize the topography of the ERP and HG activities, we used the Brainstorm (*Tadel et al., 2011*) package of MATLAB to map the results on the ICBM152 cortical surface. It is a simple 3D interpolation with Shepard's method (weights decreasing with the square of the distance), similar to what is done when projecting subject sources to an anatomy template, where vertices of the cortical surface that are further than 15 mm from the center of the sEEG contact are ignored.

For ERP visualization, we normalized the activity in absolute value because it is not clearly understood so far regarding the biological characteristics of LFP polarity. To simplify such complex issue, we consider the change in magnitude of LFP during delay period in our task represents awareness related activity, regardless its actual value being positive or negative. Therefore, we first calculated the absolute value of activity difference between aware and unaware trials in individual recording site, then used Shepard's method to calculate the activity in each vertex and projected on the surface of brain template as shown in *Figure 3B*.

## Decoding Analysis

We implemented a machine learning framework for trial-by-trial classification using broadband amplitude and high-gamma magnitude of LFP activities in the PFC via the MVPA-Light toolbox (*Guggenmos et al., 2018*). We applied the LDA classification and temporal generalization on each participant separately, and all trials from each class were undersampled to balance the unequal trial number between different conditions. Then, the trials were randomly assigned into 10 folds, and all trials within each fold were subaveraged across every four trials to increase the signal-to-noise ratio. Decoding was then followed with a leave-one-out cross-validation procedure on the subaveraged trials. This procedure was repeated 10 times so that each fold in the dataset was used exactly once for testing, and at least once for training, but never at the same time. And the fold assignment was repeated five times, which resulted in total 50 iterations for the decoding analysis of each patient.

## Functional Connectivity analysis

The time-across phase-locking value (PLV) was calculated in the brainstorm toolbox. Time-frequency transformation was conducted via the Hilbert transform method. The time-across PLV measures if the phase difference between two signals is consistent across trials with respect to the event that defines the trials (i.e. the output is a mean across trials per time point) (*Mercier et al., 2022*). PLV values range between 0 (for totally random) and 1 (for perfect phase locking). For clear demonstration here, the mean PLV in the baseline period (−200–0ms) of each pair of recording sites was subtracted. The group analysis of PLV was performed according to *Betzel et al., 2019*; *Sadaghiani et al., 2022*. The topographic display of the connectivity was conducted by the BrainNet Viewer toolbox (*Xia et al., 2013*).

## Acknowledgements

We thank the participants for volunteering to take part in the study. This study was funded by STI2030-Major Projects +2021ZD0204300. National Natural Science Foundation of China (32061143004, 32030045). Funded by Open Research Fund of the State Key Laboratory of Cognitive Neuroscience and Learning (CNLZD2202).

## Additional information

### Funding

| Funder | Grant reference number | Author |
|---|---|---|
| Ministry of Science and Technology of the People's Republic of China | STI2030-Major Projects+2021ZD0204300 | Mingsha Zhang |
| National Natural Science Foundation of China | 32061143004 | Mingsha Zhang |
| National Natural Science Foundation of China | 32030045 | Mingsha Zhang |
| State Key Laboratory of Cognitive Neuroscience and Learning | Open Research Fund CNLZD2202 | Yongzheng Han |

The funders had no role in study design, data collection and interpretation, or the decision to submit the work for publication.

### Author contributions

Zepeng Fang, Conceptualization, Resources, Data curation, Software, Formal analysis, Validation, Investigation, Visualization, Methodology, Writing – original draft, Writing – review and editing; Yuanyuan Dang, Resources, Data curation, Investigation, Methodology, Writing – original draft; Zhipei Ling, Resources, Investigation, Methodology, Project administration; Yongzheng Han, Data curation, Investigation, Methodology; Hulin Zhao, Xin Xu, Resources, Data curation, Supervision, Investigation, Methodology, Writing – original draft, Project administration, Writing – review and editing; Mingsha Zhang, Conceptualization, Resources, Formal analysis, Supervision, Funding acquisition, Validation, Methodology, Writing – original draft, Project administration, Writing – review and editing

### Author ORCIDs

Zepeng Fang https://orcid.org/0000-0002-0396-6463
Mingsha Zhang https://orcid.org/0000-0002-5407-7770

### Ethics

All subjects provided informed consent to participate in this study. The Ethics Committee of Chinese PLA General Hospital approved the experimental procedures (approval numbers S2022-457-01).

Reviewer #1 (Public Review): https://doi.org/10.7554/eLife.89076.3.sa1
Reviewer #2 (Public Review): https://doi.org/10.7554/eLife.89076.3.sa2

Reviewer #3 (Public Review): https://doi.org/10.7554/eLife.89076.3.sa3
Author Response https://doi.org/10.7554/eLife.89076.3.sa4

## Additional files

### Supplementary files
• MDAR checklist

### Data availability

The raw data, including behavioral data, sEEG data, the deidentified brain imaging data, are available from the Dryad Digital Repository: http://dx.doi.org/10.5061/dryad.p8cz8w9xp. The code for data analysis is uploaded as source code files.

The following dataset was generated:

| Author(s) | Year | Dataset title | Dataset URL | Database and Identifier |
|---|---|---|---|---|
| Fang Z, Dang Y, Ling Z, Han Y, Zhao H, Xu X, Zhang M | 2024 | The raw data, including behavioral data, sEEG data, the deidentified brain imaging data | https://doi.org/10.5061/dryad.p8cz8w9xp | Dryad Digital Repository, 10.5061/dryad.p8cz8w9xp |

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
