## [Editor Report · eLife assessment]

This paper reports **valuable** results regarding the potential role and time course of the prefrontal cortex in conscious perception. Although the sample size is small, the results are **convincing**, and strengths include the use of several complementary analysis methods. The behavioral test includes subject report such that the study does not allow for distinguishing between (phenomenal) awareness and conscious access; nevertheless, results do advance our understanding of the contribution of prefrontal cortex to conscious perception.

---

## [Referee Report · Reviewer #1 (Public Review)]

This is a clear and rigorous study of intracranial EEG signals in the prefrontal cortex during a visual awareness task. The results are convincing and worthwhile, and strengths include the use of several complementary analysis methods and clear results. The only methodological weakness is relatively small sample size of only 6 participants compared to other studies in the field. Interpretation weaknesses are claims that their task removes the confound of report (it does not), and claims of primacy in showing early prefrontal cortical involvement in visual perception using intracranial EEG (several studies already have shown this). Also the shorter reaction times for perceived vs not perceived stimuli (confident vs not confident responses) has been described many times previously and is not a new result.

---

## [Referee Report · Reviewer #2 (Public Review)]

The authors attempt to address a long-standing controversy in the study of the neural correlates of visual awareness, namely whether neurons in prefrontal cortex are necessarily involved in conscious perception. Several leading theories of consciousness propose a necessary role for (at least some sub-regions of) PFC in basic perceptual awareness (e.g., global neuronal workspace theory, higher order theories), while several other leading theories posit that much of the previously reported PFC contributions to perceptual awareness may have been confounded by task-based cognition that co-varied between the aware and unaware reports (e.g., recurrent processing theory, integrated information theory). By employing intracranial EEG in human patients and a threshold detection task on low-contrast visual stimuli, the authors assessed the timing and location of neural populations in PFC that are differentially activated by stimuli that are consciously perceived vs. not perceived. Overall, the reported results support the view that certain regions of PFC do contribute to visual awareness, but at time-points earlier than traditionally predicted by GNWT and HOTs.

Major strengths of this paper include the straightforward visual threshold detection task including the careful calibration of the stimuli and the separate set of healthy control subjects used for validation of the behavioral and eye tracking results, the high quality of the neural data in six epilepsy patients, the clear patterns of differential high gamma activity and temporal generalization of decoding for seen versus unseen stimuli, and the authors' interpretation of these results within the larger research literature on this topic. This study appears to have been carefully conducted, the data were analyzed appropriately, and the overall conclusions seem warranted given the main patterns of results.

Weaknesses include the saccadic reaction time results and the potential flaws in the design of the reporting task. As the authors acknowledge, this is not a "no report" paradigm, rather, it's a paradigm aimed at balancing the post-perceptual cognitive and motor requirements between the seen and unseen trials. On each trial, subjects/patients either perceived the stimulus or not, and had to briefly maintain this "yes/no" judgment until a fixation cross changed color, and the color change indicated how to respond (saccade to the left or right). Differences in saccadic RTs (measured from the time of the fixation color change to moving the eyes to the left or right response square) were evident between the seen and unseen trials (faster for seen). In the discussion, the authors summarize several alternative explanations of the saccade results and limitations of their report paradigm that will help guide future research.

The current results help advance our understanding of the contribution of PFC to visual awareness. These results, when situated within the larger context of the rapidly developing literature on this topic provide converging evidence that some sub-regions of PFC contribute to visual awareness, but at latencies earlier than originally predicted by proponents of, especially, global neuronal workspace theory. Three recent studies that used "no report paradigms", but with clearly visible stimuli, reported very similar results in PFC (Vishne et al., 2023; Broday-Dvir et al., 2023; Cogitate et al., 2023). The current study uses a report paradigm, but with consciously seen vs. unseen conditions, to fill the gap left by these previous studies, i.e., it remained unclear whether the PFC results from the previous studies were related to conscious or unconscious processing. Taken as a whole, evidence appears to be converging for a limited and early-in-time (200-300ms) contribution of PFC to visual awareness, after task and motor confounds are minimized.

---

## [Referee Report · Reviewer #3 (Public Review)]

The authors report a study in which they use intracranial recordings to dissociate subjectively aware and subjectively unaware stimuli, focusing mainly on prefrontal cortex.

The authors have dealt successfully with some of my previous concerns, especially the more direct link to the Gaillard et al., (2009) paper, and the associated analyses, has improved the manuscript. Some of my other concerns regarding the theoretical embedding of the findings have only been partially mitigated and some interesting results derived from suggestions for additional analyses will be used for future papers.

---

## [Author Response]

The following is the authors’ response to the original reviews.

**eLife assessment**
This paper reports valuable results regarding the potential role and time course of the prefrontal cortex in conscious perception. Although the sample size is small, the results are clear and convincing, and strengths include the use of several complementary analysis methods. The behavioral test includes subject report so the results do not allow for distinguishing between theories of consciousness; nevertheless, results do advance our understanding of the contribution of prefrontal cortex to conscious perception.We appreciate very much for editor and reviewers encouraged review opinion. Particularly, we thank three reviewers very much for their professional and constructive comments that help us to improve the manuscript substantially.
**Public Reviews:**

**Reviewer #1 (Public Review):**
This is a clear and rigorous study of intracranial EEG signals in the prefrontal cortex during a visual awareness task. The results are convincing and worthwhile, and strengths include the use of several complementary analysis methods and clear results. The only methodological weakness is the relatively small sample size of only 6 participants compared to other studies in the field. Interpretation weaknesses that can easily be addressed are claims that their task removes the confound of report (it does not), and claims of primacy in showing early prefrontal cortical involvement in visual perception using intracranial EEG (several studies already have shown this). Also the shorter reaction times for perceived vs not perceived stimuli (confident vs not confident responses) has been described many times previously and is not a new result.

We appreciate very much for the reviewer’s encouraged opinion. We are going to address reviewer’s specific questions and comments point-by-point in following.

‘The only methodological weakness is the relatively small sample size of only 6 participants compared to other studies in the field.’

We agree that the sample size is relatively small in the present study. To compensate such shortcoming, we rigorously verified each result at both individual and population levels, resembling the data analysis method in non-human primate study.

Interpretation weaknesses that can easily be addressed are claims that their task removes the confound of report (it does not),

Thank you very much for your comment. We agree that our task does not remove the confound of report entirely. However, we believe that our task minimizes the motor confounds by dissociating the emergence of awareness from motor in time and balanced direction of motor between aware and unaware conditions. We have modified the text according to reviewer’s comment in the revised manuscript as following: “This task removes the confound of motor-related activity”.

..and claims of primacy in showing early prefrontal cortical involvement in visual perception using intracranial EEG (several studies already have shown this).

We agree that several iEEG studies, including ERP and HFA, have shown the early involvement of prefrontal cortical in visual perception. However, in these studies, the differential activity between conscious and unconscious conditions was not investigated, thus, the activity in prefrontal cortex might be correlated with unconscious processing, rather than conscious processing. In present study, we compared the neural activity in PFC between conscious and unconscious trials, and found the correlation between PFC activity and conscious perception. Although one iEEG study(Gaillard et al., 2009) reported awareness-specific PFC activation, the awareness-related activity started 300 ms after the onset of visual stimuli, which was ~100 ms later than the early awareness related activity in our study. Also, due to the limited number of electrodes in the previous study (2 patients with 19 recording sites mostly in mesiofrontal and peri-insular regions), it was restricted while exploring the awareness-related activity in PFC. In the present study, the number of recording sites (245) were much more than previous study and covered multiple areas in PFC. Our results further show earlier awareness-related activity (~ 200 ms after visual stimuli onset), including ERP, HFA and PLV, which sheds new light on understanding of the role of PFC in conscious perception.

We have added this discussion in the MS (lines 522-536);

Also the shorter reaction times for perceived vs not perceived stimuli (confident vs not confident responses) has been described many times previously and is not a new result.Thank you very much for your comment. We agree that the reaction time is strongly modulated by the confident level, which has been described previously (Broggin, Savazzi, & Marzi, 2012; Marzi, Mancini, Metitieri, & Savazzi, 2006). However, in previous studies, the confident levels were usually induced by presenting stimulus with different physical property, such as spatial frequency, eccentricity and contrast. It is well known that the more salient stimuli will induce the faster process of visual information and speed up the process of visuomotor transformation, eventually shorten the reaction time (Corbetta & Shulman, 2002; Posner & Petersen, 1990). Therefore, the dependence of visual processing on the salience of visual stimulus confounds with the effect of visual awareness on the reaction time, which is hard to attribute the shorter reaction time in more salient condition purely to visual awareness. In contrast, we create a condition (near perceptual threshold) in the present study, in which the saliency (contrast) of visual stimulus is very similar in both aware and unaware conditions in order to eliminate the influence of stimulus saliency in reaction time. We think that the difference in reaction time in our study is mainly due to the modulation of awareness state, which was not reported previously.

We have added the discussion in the MS (lines 497-507).

**Reviewer #1 (Recommendations For The Authors):**
Specific comments follow:Abstract: "we designed a visual awareness task that can minimize report-related confounding" and in the Introduction lines 112-115: "Such a paradigm can effectively dissociate awareness-related activity from report-related activity in terms of time... and report behavior"; Discussion lines 481-483 "even after eliminating the influence of the confounding variables related to subjective reports such as motion preparation" and other similar statements in the manuscript should be removed. The task involves report using eye movements with every single stimulus. The fact that there is report for both perceived and not perceived stimuli, that the direction of report is not determined until the time of report, and that there is delay between stimulus and report, does not remove the report-related post-perceptual processing that will inevitably occur in a task where overt report is required for every single trial. For example, brain activity related to planning to report perception will only occur after perceived trials, regardless of the direction of eye movement later decided upon. This preparation to respond is different for perceived and not perceived stimuli, but is not part of the perception itself. In this way the current task is not at all unique and does not substantially differ from many other report-based tasks used previously.

The objective of present study is to assess whether PFC is involved in the emergence of visual awareness. To do so, it is crucial to determine the subjective awareness state as correct as possible. Considering the disadvantage of non-report paradigms in determining the subjective awareness state (Tsuchiya et al. TiCS, 2015; Mashour et al, Neuron, 2020), we employed a balanced report paradigm. It has been argued (Merten & Nieder, PNAS, 2011) that, in the balanced report paradigms, subjects could not prepare any motor response during the delay period because only the appearance of a rule cue (change color of fixation point at the end of delay period) informed subjects about the appropriate motor action. In this case, the post-perceptual processing during delay period might reflect the non-motor cognitive activity. Alternatively, as being mentioned by reviewer, the post-perceptual processing might relate to planning to report perception, which is different for perceived and not perceived stimuli. Therefore, up to date, the understanding of the post-perceptual processing remains controversial. According to reviewer’s comment, we have modified the description of our task as following: “we designed a visual awareness task that can minimize report-related motor confounding”. Also, have changed “report-related” to “motorrelated” in the text of manuscript.

Figures 3, 4 changes in posterior middle frontal gyri suggest early frontal eye field involvement in perception. This should be interpreted in the context of many previous studies showing FEF involvement in signal detection. The authors claim that "earlier visual awareness related activities in the prefrontal cortex were not found in previous iEEG studies, especially in the HG band" on lines 501-502 of the Discussion. This statement is not true and should be removed. The following statement in the Discussion on lines 563-564 should be removed for the same reasons: "our study detected 'ignition' in the human PFC for the first time." Authors should review and cite the following studies as precedent among others:Blanke O, Morand S, Thut G, Michel CM, Spinelli L, Landis T, Seeck M (1999) Visual activity in the human frontal eye field. Neuroreport 10 (5):925-930. doi:10.1097/00001756-19990406000006Foxe JJ, Simpson GV (2002) Flow of activation from V1 to frontal cortex in humans. A framework for defining "early" visual processing. Exp Brain Res 142 (1):139-150. doi:10.1007/s00221-001-0906-7Gaillard R, Dehaene S, Adam C, Clemenceau S, Hasboun D, Baulac M, Cohen L, Naccache L (2009) Converging intracranial markers of conscious access. Plos Biology 7 (3):e61Gregoriou GG, Gotts SJ, Zhou H, Desimone R (2009) High-frequency, long-range coupling between prefrontal and visual cortex during attention. Science 324:1207-1210Herman WX, Smith RE, Kronemer SI, Watsky RE, Chen WC, Gober LM, Touloumes GJ, Khosla M, Raja A, Horien CL, Morse EC, Botta KL, Hirsch LJ, Alkawadri R, Gerrard JL, Spencer DD, Blumenfeld H (2019) A Switch and Wave of Neuronal Activity in the Cerebral Cortex During the First Second of Conscious Perception. Cereb Cortex 29 (2):461-474.Khalaf A, Kronemer SI, Christison-Lagay K, Kwon H, Li J, Wu K, Blumenfeld H (2022) Early neural activity changes associated with stimulus detection during visual conscious perception. Cereb Cortex. doi:10.1093/cercor/bhac140Kwon H, Kronemer SI, Christison-Lagay KL, Khalaf A, Li J, Ding JZ, Freedman NC,Blumenfeld H (2021) Early cortical signals in visual stimulus detection. Neuroimage 244:118608.

We agree that several iEEG studies, including ERP and HFA, have shown the early involvement of prefrontal cortical in visual perception. However, in these studies, the differential activity between conscious and unconscious conditions was not investigated, thus, the activity in prefrontal cortex might be correlated with unconscious processing, rather than conscious processing. In present study, we compared the neural activity in PFC between conscious and unconscious trials, and found the correlation between PFC activity and conscious perception. Although one iEEG study reported awareness-specific PFC activation, the awareness-related activity started 300 ms after the onset of visual stimuli, which was ~100 ms later than the early awareness related activity in our study. Also, due to the limited number of electrodes in the previous study (2 patients with 19 recording sites mostly in mesiofrontal and peri-insular regions), it was restricted while exploring the awareness-related activity in PFC. In the present study, the number of recording sites (245) were much more than previous study and covered multiple areas in PFC. Our results further show earlier awareness-related activity (~ 200 ms after visual stimuli onset), including ERP, HFA and PLV, which sheds new light on understanding of the role of PFC in conscious perception.

We have added this discussion in the MS (lines 522-533);

Minor weakness that should be mentioned in the Discussion: The intervals for the FP (fixation period) and Delay period were both fixed at 600 ms instead of randomly jittered, so that subjects likely had anticipatory activity predictably occurring with each grating and cue stimulus.

Thank you very much for your comment. We agree that subjects might have anticipatory activity during experiment. Actually, the goal for us to design the task in this way is to try to balance the effect of attention and anticipation between aware and unaware conditions. We have added this discussion in the MS (lines 467-469);

The faster reaction times for perceived/confident responses vs not perceived/unconfident responses has been reported many times previously in the literature and should be acknowledged rather than being claimed as a novel finding. Authors should modify p. 163 lines 160-162, first sentence of the Discussion lines 445-446 "reaction time.. shorter" claiming this was a novel finding; same for lines 464-467. Please see the following among others:Broggin E, Savazzi S, Marzi CA (2012) Similar effects of visual perception and imagery on simple reaction time. Q J Exp Psychol (Hove) 65 (1):151-164. doi:10.1080/17470218.2011.594896Chelazzi L, Marzi CA, Panozzo G, Pasqualini N, Tassinari G, Tomazzoli L (1988) Hemiretinal differences in speed of light detection in esotropic amblyopes. Vision Res 28 (1):95-104Marzi CA, Mancini F, Metitieri T, Savazzi S (2006) Retinal eccentricity effects on reaction time to imagined stimuli. Neuropsychologia 44 (8):1489-1495. doi:10.1016/j.neuropsychologia.2005.11.012Posner MI (1994) Attention: the mechanisms of consciousness. Proceedings of the National Academy of Sciences of the United States of America 91 (16):7398-7403Sternberg S (1969) Memory-scanning: mental processes revealed by reaction-time experiments. Am Sci 57 (4):421-457

Thanks. We have cited some of these papers in the revised manuscript due to the restricted number of citations.

Methods lines 658-659: "results under LU and HA conditions were classified as the control group and were only used to verify and check the results during calculation." However the authors show these results in the figures and they are interesting. HA stimuli show earlier responses than NA stimuli. This is a valuable result which should be discussed and interpreted in light of the other findings.

We thank very much for reviewer’s comment. We have made discussion accordingly in the revised MS (lines 535-536).

General comment on figures: Many of the figure elements are tiny and the text labels and details can't be seen at all, especially single trial color plots, and the brain insets showing recording sites.

We have modified the figures accordingly.

Other minor comments:Typo: Figure 2 legend, line 169 "The contrast level resulted in an awareness percentage greater than 25%..." is missing a word and should say instead something like "The contrast level that resulted in an awareness percentage greater than 25%..."

Thanks. We have corrected the typo accordingly.

Figure 2 Table description in text line 190 says "proportions of recording sites" but the Table only shows number of recording sites and number of subjects, not "proportions." This should be corrected in the text.

Thanks. We have corrected the error.

Figure 3, and other figures, should always label the left and right hemispheres to avoid ambiguity.

Thanks. We have made correction accordingly. In caption of Figure 2D (line 189), we modified the sentence as ‘In all brain images, right side of the image represents the right side of the brain’.

Methods line 666. The saccadic latency calculations paragraph should have a separate heading before it, to separate it from the Behavioral data analysis section.

Thanks. It has been corrected in line 725.

**Reviewer #2 (Public Review):**
The authors attempt to address a long-standing controversy in the study of the neural correlates of visual awareness, namely whether neurons in prefrontal cortex are necessarily involved in conscious perception. Several leading theories of consciousness propose a necessary role for (at least some sub-regions of) PFC in basic perceptual awareness (e.g., global neuronal workspace theory, higher order theories), while several other leading theories posit that much of the previously reported PFC contributions to perceptual awareness may have been confounded by task-based cognition that co-varied between the aware and unaware reports (e.g., recurrent processing theory, integrated information theory). By employing intracranial EEG in human patients and a threshold detection task on low-contrast visual stimuli, the authors assessed the timing and location of neural populations in PFC that are differentially activated by stimuli that are consciously perceived vs. not perceived. Overall, the reported results support the view that certain regions of PFC do contribute to visual awareness, but at time-points earlier than traditionally predicted by GNWT and HOTs.

Reply: We appreciate very much for the reviewer’s encouraged opinion.

Major strengths of this paper include the straightforward visual threshold detection task including the careful calibration of the stimuli and the separate set of healthy control subjects used for validation of the behavioral and eye tracking results, the high quality of the neural data in six epilepsy patients, the clear patterns of differential high gamma activity and temporal generalization of decoding for seen versus unseen stimuli, and the authors' interpretation of these results within the larger research literature on this topic. This study appears to have been carefully conducted, the data were analyzed appropriately, and the overall conclusions seem warranted given the main patterns of results.

Reply: We appreciate very much for the reviewer’s encouraged opinion.

Weaknesses include the saccadic reaction time results and the potential flaws in the design of the reporting task. This is not a "no report" paradigm, rather, it's a paradigm aimed at balancing the post-perceptual cognitive and motor requirements between the seen and unseen trials. On each trial, subjects/patients either perceived the stimulus or not, and had to briefly maintain this "yes/no" judgment until a fixation cross changed color, and the color change indicated how to respond (saccade to the left or right). Differences in saccadic RTs (measured from the time of the fixation color change to moving the eyes to the left or right response square) were evident between the seen and unseen trials (faster for seen). If the authors' design achieved what they claim on page 3, "the report behaviors were matched between the two awareness states ", then shouldn't we expect no differences in saccadic RTs between the aware and unaware conditions? The fact that there were such differences may indicate differences in post-perceptual cognition during the time between the stimulus and the response cue. Alternatively, the RT difference could reflect task-strategies used by subjects/patients to remember the response mapping rules between the perception and the color cue (e.g., if the YES+GREEN=RIGHT and YES+RED=LEFT rules were held in memory, while the NO mappings were inferred secondarily rather than being actively held in memory). This saccadic RT result should be better explained in the context of the goals of this particular reporting-task.

The objective of present study is to assess whether PFC is involved in the emergence of visual awareness. To do so, it is crucial to determine the subjective awareness state as correct as possible. Considering the disadvantage of non-report paradigms in determining the subjective awareness state (Tsuchiya et al, TiCS, 2015; Mashour et al, Neuron, 2020), we employed a balanced report paradigm. It has been argued (Merten & Nieder, PNAS, 2011) that, in the balanced report paradigms, subjects could not prepare any motor response during the delay period because only after the appearance of a rule cue (change color of fixation point at the end of delay period) subjects were informed about the appropriate motor action. In this case, the post-perceptual processing during delay period might reflect the non-motor cognitive activity, such as working memory (Mashour et al. Neuron, 2020). Alternatively, as being mentioned by reviewer, the postperceptual processing might relate to planning to report perception, which is different for perceived and not perceived stimuli (Aru et al. Neurosci Biobehav Rev, 2012 ). Therefore, up to date, the understanding of the post-perceptual processing remains controversial. Considering reviewer’s comment together with other opinions, we have modified the description of our task as following: “we designed a visual awareness task that can minimize report-related motor confounding”. Also, we have changed “report-related” to “motor-related” in the rest of manuscript.

Regarding the question whether the saccadic RT in our balanced response paradigm should be expected to be similar between aware and unaware condition, we think that the RT should be similar in case if the delay period is long enough for the decision of “no” to be completed. In fact, in a previous study (Merten & Nieder, PNAS, 2011), the neuronal encoding of “no” decision didn’t appear until 2s after the stimulus cue onset. However, in our task, the delay period lasted only 600 ms that was long enough to form the “yes” decision, but was not enough to form the “no” decision. It might be the reason that our data show shorter RT in aware condition than in unaware condition.

We totally agree reviewer’s comment about the alternative interpretation for RT difference between aware and unaware condition in our study, i.e., reflecting task-strategies used by subjects/patients to remember the response mapping rules between the perception and the color cue (e.g., if the YES+GREEN=RIGHT and YES+RED=LEFT rules were held in memory, while the NO mappings were inferred secondarily rather than being actively held in memory).We have made additional discussion about these questions in the revised manuscript (lines 492496).

Nevertheless, the current results do help advance our understanding of the contribution of PFC to visual awareness. These results, when situated within the larger context of the rapidly developing literature on this topic (using "no report" paradigms), e.g., the recent studies by Vishne et al. (2023) Cell Reports and the Cogitate consortium (2023) bioRxiv, provide converging evidence that some sub-regions of PFC contribute to visual awareness, but at latencies earlier than originally predicted by proponents of, especially, global neuronal workspace theory.

We appreciate very much for the reviewer’s encouraged opinion.

**Reviewer #2 (Recommendations For The Authors):**
Abstract: "the spatiotemporal overlap between the awareness-related activity and the interregional connectivity in PFC suggested that conscious access and phenomenal awareness may be closely coupled." I strongly suggest revising this sentence. The current results cannot be used to make such a broad claim about p-consciousness vs. a-consciousness. This study used a balanced trial-by-trial report paradigm, which can only measure conscious access.

We thank reviewer for this comment. We have withdrawn this sentence from the revised manuscript.

Task design: A very similar task was used previously by Schröder et al. (2021) J Neurosci. See specifically, their Figure 1, and Figure 4B-C. Using almost the exact same "matching task", the authors of this previous study show that they get a P3b for both the perceived and not-perceived conditions, confirming that post-perceptual cognition/report confounds were not eliminated, but instead were present in (and balanced between) both the perceived/not-perceived trials due to the delayed matching aspect of the design. This previous paper should be cited and the P3b result should be considered when assessing whether cognition/report confounds were addressed in the current study.

Thank you very much for your reminding about the study of Schröder et al. We are sorry for not citing this closely related study in our previous manuscript. Schröder et al. found while P3b showed significant difference between perceived and not-perceived trials in direct report task, the P3b was presented in both perceived/not-perceived trials and not significantly different in the matched task. Based on these findings, Schröder et al. argued that P3b represented the task specific post-perceptual cognition/report rather than the emergence of awareness per se. Considering the similarity of tasks between Schröder et al. and ours, we agree that our task is not able to totally eliminate the confound of post-perceptual cognition/report related activity with awareness related activity. Nevertheless, our task is able to minimize the confound of motorrelated activity with the emergence of awareness by separating them in time and balancing the direction of responsive movements. Therefore, we modified the term of “report-related” to “motor-related” in the text of revised manuscript.

On page 2, lines 71-75, the authors' review of the Frassle et al. (2014) experiment should be revised for accuracy. In this study, all PFC activity did not disappear as the authors claim. Also, the main contrast in the Frassle et al. study was rivalry vs. replay. However, in both of these conditions, visual awareness was changing with the main difference being whether there was sensory conflict between the two eyes or not. Such a contrast would presumably subtract out the common activity patterns related to visual awareness changes, while isolating rivalry (and the resulting neural competition) vs. non-rivalry (and the lack of such competition) which is not broadly relevant for the goal of measuring neural correlates of visual awareness which are present in both sides of the contrast (rivalry and replay).

Thank you very much for your suggestion. We agree that and revised in the MS (lines 71-76).

‘For instance, a functional magnetic resonance imaging (fMRI) study employing human binocular rivalry paradigms found that when subjects need to manually report the changing of their awareness between conflict visual stimuli, the frontal, parietal, and occipital lobes all exhibited awareness-related activity. However, when report was not required, awareness-related activation was largely diminished in the frontal lobe but remained in the occipital and parietal lobes’

On page 2, lines 76-78, the authors write, "no-report paradigm may overestimate unconscious processing because it cannot directly measure the awareness state". This should be reworded for clarity, as report paradigms also do not "directly measure the awareness state". All measures of awareness are indirect, either via subjects verbal or manual reports, or via behaviors or other physiological measures like OKN, pupillometry, etc. It's also not clear as written why no-report paradigms might overestimate unconscious processing.

Thank you very much for your suggestion. We agreed and modified the description.In lines 76-80:

‘Nevertheless, the no-report paradigm may overestimate the neural correlates of awareness by including unconscious processing, because it infers the awareness state through other relevant physiological indicators, such as optokinetic nystagmus and pupil size(Tsuchiya, Wilke, Frassle, & Lamme, 2015). In the absence of subjective reports, it remains controversial regarding whether the presented stimuli are truly seen or not.’

However, the no-report paradigm may overestimate the neural correlates of awareness, because it infers the awareness state through other relevant physiological indicators, such as optokinetic nystagmus and pupil size(Tsuchiya et al., 2015) , in the absence of subjective reports and it remains controversial that whether the stimuli presented in such paradigm are truly seen as opposed to being merely potentially visible but unattended.On page 5, line 155, there is a typo. This should be Figure 2C, not 2B.

Thanks. We have modified the description.

On page 5, lines 160-162, the authors state, "The results showed that the saccadic reaction time in the aware trials was systematically shorter than that in the unaware trials. Such results demonstrate that visual awareness significantly affects the speed of information processing in the brain." I don't understand this. If subjects can never make a saccade until the fixation cross changes color, both for Y and N decisions, why would a difference in saccadic reaction times indicate anything about visual awareness affecting the speed of information processing in the brain? Doesn't this just show that the Red/Green x Left/Right response contingencies were easier to remember and execute for the Yes-I-did-see-it decisions compared to the No-I-didn't-see-it decisions?

We agree and have made additional discussion about these questions in the revised manuscript (lines 492-496).

‘An alternative interpretation for RT difference between aware and unaware condition in our study is that the difference in task-strategies used by subjects/patients to remember the response mapping rules between the perception and the color cue (e.g., if the YES+GREEN=RIGHT and YES+RED=LEFT rules were held in memory, while the NO mappings were inferred secondarily rather than being actively held in memory).’

In Figure 3B (and several other figures) due to the chosen view and particular brain visualization used, many readers will not know whether the front of brain is up and back of brain down or vise versa (there are no obvious landmarks like the cerebellum, temporal sulcus, etc.). I suggest specifying this in the caption or better yet on the figure itself.

Thanks. We have added these descriptions in the caption of Figure 2D.

Line 189 ‘In all brain images, right and up sides of each image represent the right and up sides of the brain’.In Figure 3B, the color scale may confuse some readers. When I first inspected this figure, I immediately thought the red meant positive voltage or activation, while the blue meant negative voltage or deactivation. Only later, I realized that any color here is meaningful. Not sure if an adjustment of the color scale might help, or perhaps not normalizing (and not taking absolute values of the voltage diffs, but maintaining the +/- diffs)?

Thanks for reviewer’s comment. We are sorry for not clearly describing the reason why we normalized the activity in absolute value and chose the color scale from 0 to 20. The major reason is that it is not clearly understood so far regarding the biological characteristics of LFP polarity (Einevoll et al, Nat Rev Neurosci, 2013). To simplify such complex issue, we consider the change in magnitude of LFP during delay period in our task represents awareness related activity, regardless its actual value being positive or negative. Therefore, we first calculated the absolute value of activity difference between aware and unaware trials in individual recording site, then used Shepard's method (see Method for detailed information) to calculate the activity in each vertex and projected on the surface of brain template as shown in Fig. 3B.

We have added the description in the MS (lines 794-800).

We have tried to adjust the color scale from -20 to 20 according to reviewer’s suggestion. However, the topographic heatmap showed less distinguishable between brain regions with different strength of awareness related activity. Thus, we would like to keep the way as we used to analyze and present these results.

Figure 3B: Why choose seemingly arbitrary time points in this figure? What's the significance of 247 and 314 and 381ms (why not show 200, 250, 300, etc.)? Also, are these single time-points or averages within a broader time window around this time-point, e.g., 225-275ms for the 250ms plot?

Thank reviewer for this helpful comment. We are sorry for not clearly describing why we chose the 8 time points to demonstrate the spatiotemporal characteristics of awareness related activity in Fig. 3B. To identify the awareness related activity, we analyzed the activity difference between aware and unaware trials during delay period (180-650 ms after visual stimulus onset). The whole dynamic process has been presented in SI with a video (video S1). Here, we just sampled the activity at 8 time points (180 ms, 247 ms, 314 ms, etc.) that equally divided the 430 ms delay period.

We have added the description in the MS (lines 213-215).

Figure 3D: It's not clear how this figure panel is related to the data shown in Fig3A. In Fig3A, the positive amplitude diffs all end at around 400ms, but in Fig3D, these diffs extend out to 600+ms. I suggest adding clarity about the conversion being used here.

Thanks for reviewer’s comment. We are sorry for not clearly describing the way to analyze the population activity (Fig. 3D) in the previous version of manuscript. Since it is not clearly understood so far regarding the biological characteristics of LFP polarity, to simplify such complex issue, we consider the change in magnitude of LFP during delay period in our task is awareness related activity, regardless its actual value being positive or negative. Therefore, while analyzing the awareness related population activity, we first calculate the absolute value of activity difference between aware and unaware trials in individual recording site, then pool the data of 43 recording sites together and calculate the mean and standard error of mean (SEM)(Fig. 3D). As you can see in Fig. 3A, the activity difference between aware (red) and unaware (blue) trials lasts until/after the end of delay period. Thus, the awareness related population activity in Fig 3D extends out to 600 ms.

We have added the description in the MS (lines 769-777).

Figure 6D could be improved by making the time labels much bigger, perhaps putting them on the time axis on the bottom rather than in tiny text above each brain.

Thanks for reviewer’s comment. We have modified it accordingly.

Page 18, line 480: "our results show that the prefrontal cortex still displays visual awareness-related activities even after eliminating the influence of the confounding variables related to subjective reports such as motion preparation" This is too strong of a statement. It's not at all clear whether confounding variables related to subjective reports (especially the cognition needed to hold in mind the Y/N decision about seeing the stimulus prior to the response cue) were eliminated with the design used here. In other places of the manuscript, the authors use "minimized" which is more accurate.

Thanks for reviewer’s comment. We have modified it accordingly.

Page 19, section starting on line 508: The authors should consider citing the study by Vishne et al. (2023), which was just accepted for publication recently, but has been posted on bioRxiv for almost a year now: https://www.biorxiv.org/content/10.1101/2022.08.02.502469v1 . And on page 20, line 563, the authors claim that to the best of their knowledge, they were the first to detect "ignition" in PFC in human subjects. Consider revising this statement, now that you know about the Vishne et al. paper.

We agree.

Thanks for your reminding about these papers. We have cited this study and made discussion in the revised manuscript (line 522-533). We agree that several iEEG studies have shown the early involvement of PFC in visual perception (Vishne et al. 2023; Khalaf et al. 2023; Kwon et al. 2021). However, in these studies, authors did not compare the neural activity between conscious and unconscious conditions, leaving the possibility that the ERP and HFA were correlated with the unconscious information processing rather than awareness-specific processing. In the present study, we compared the neural activity in PFC between conscious and unconscious trials, and found that the activity of PFC specifically correlated with conscious perception. As we mentioned in the previous version of manuscript, there is one iEEG study (Gaillard et al. 2009) that reported awareness-specific activity in PFC. However, the awareness related activity started more than 300 ms after the onset of visual stimuli, which was about 100 ms longer than the early awareness related activity in our study. Nevertheless, according to reviewer’s comment, we modified our argument as following in lines 621-623:

‘However, as discussed above, in contrast with previous studies, our study detected earlier awareness-specific ‘ignition’ in the human PFC, while minimizing the motor-related confounding.’

Experimental task section of Methods: Were any strategies for learning the response cue matching task suggested to patients/subjects, and/or did any patients/subjects report which strategy they ended up using? For example, if I were a subject in this experiment, I would remember and mentally rehearse the rules: "YES+GREEN = RIGHT" and "YES+RED = LEFT". For trials in which I didn't see anything, I wouldn't need to hold 2 more rules in mind, as they can be inferred from the inverse of the YES rules (and it's much harder to hold 4 things in mind than 2). This extra inference needed to get to the NO+GREEN = LEFT and NO+RED = RIGHT rules would likely cause me to respond slightly slower to the NO trials compared to the YES trials, leading to saccadic RT effects in the same direction the authors found. More information about the task training and strategies used by patients/subjects would be helpful.

We agree and discussed this in lines 492-496.

**Reviewer #3 (Public Review):**
The authors report a study in which they use intracranial recordings to dissociate subjectively aware and subjectively unaware stimuli, focusing mainly on prefrontal cortex. Although this paper reports some interesting findings (the videos are very nice and informative!) the interpretation of the data is unfortunately problematic for several reasons. I will detail my main comments below. If the authors address these comments well, I believe the paper may provide an interesting contribution to further specifying the neural mechanisms important for conscious access (in line with Gaillard et al., Plos Biology 2009).

Reply: We appreciate very much for the reviewer’s encouraged opinion.

The main problem with the interpretation of the data is that the authors have NOT used a so called "no-report paradigm". The idea of no report paradigms is that subjects passively view a certain stimulus without the instruction to "do something with it", e.g., detect the stimulus, immediately or later in time. Because of the confusion of this term, specifically being related to the "act of reporting", some have argued we should use the term no-cognition paradigm instead (Block, TiCS, 2019, see also Pitts et al., Phil Trans B 2018). The crucial aspect is that, in these types of paradigms, the critical stimulus should be task-irrelevant and thus not be associated with any task (immediately or later). Because in this experiment subjects were instructed to detect the gratings when cued 600 ms later in time, the stimuli are task relevant, they have to be reported about later and therefore trigger all kinds of (known and potentially unknown) cognitive processes at the moment the stimuli are detected in real-time (so stimulus-locked). You could argue that the setup of this delayed response task excludes some very specific report related processes (e.g., the preparation of an eye-movement), which is good, however this is usually not considered the main issue. For example when comparing masked versus unmasked stimuli (Gaillard et al., 2009 Plos Biology), these conditions usually also both contain responses but these response related processes are "averaged out" in the specific contrasts (unmasked > masked). In this paper, RT differences between conditions (that are present in this dataset) are taken care of by using this delayed response in this paper, which is a nice feature for that and is not the case for the above example set-up.Given the task instructions, and this being merely a delayed-response task, it is to be expected that prefrontal cortex shows stronger activity for subjectively aware versus subjectively unaware stimuli. Unfortunately, given the nature of this task, the novelty of the findings is severely reduced. The authors cannot claim that prefrontal cortex is associated with "visual awareness", or what people have called phenomenal consciousness (this is the goal of using no-cognition paradigms). The only conclusion that can be drawn is that prefrontal cortex activity is associated with accessing sensory input: and hence conscious access. This less novel observation has been shown many times before and there is also little disagreement about this issue between different theories of consciousness (e.g., global workspace theory and local recurrency theories both agree on this).

We totally agree that the no-report/no-cognition paradigms contain less cognition within the post-perceptual processing than the report paradigms. We designed the balanced response task in order to minimize the motor related component from post-perceptual processing, even though this task does not eliminate the entire cognition from post-perceptual processing. Regarding reviewer’s comment that our task is not able to assess the involvement of PFC in the emergence of awareness, we have different opinion. As we mentioned in the manuscript, the findings of early awareness related activity (~200 ms) in PFC, which resemble the VAN activity in EEG studies, indicate the association of PFC with the emergence of visual awareness (phenomenal consciousness).

The best solution at this point seems to rewrite the paper entirely in light of this. My advice would be to state in the introduction that the authors investigate conscious access using iEEG and then not refer too much to no-cognition paradigm or maybe highlight some different strategies about using task-irrelevant stimuli (see Canales-Johnson et al., Plos Biology 2023; Hesse et al., eLife 2020; Hatamimajoumerd et al Curr Bio 2022; Alilovic et al., Plos Biology 2023; Pitts et al., Frontiers 2014; Dwarakanth et al., Neuron 2023 and more). Obviously, the authors should then also not claim that their results solve debates about theories regarding visual awareness (in the "no-cognition" sense, or phenomenal consciousness), for example in relation to the debate about the "front or the back of the brain", because the data do not inform that discussion. Basically, the authors can just discuss their results in detail (related to timing, frequency, synchronization etc) and relate the different signatures that they have observed to conscious access.

The objective of present study is to assess whether PFC is involved in the emergence of visual awareness (i.e., phenomenal consciousness). Interestingly, we found the early awareness related activity (~200 ms after visual stimulus onset), including ERP, high gamma activity and phase synchronization, in PFC, which indicate the association of PFC with the emergence of visual awareness. Therefore, we would like to keep the basic context of manuscript and make revision according to reviewers’ comments.

On the other hand, we totally agree reviewer’s argument that the report paradigm is more suitable to study the access consciousness. Indeed, we have found that the awareness related activity in PFC could be separated into two subgroups, i.e., early activity with shorter latency (~200 ms after stimulus onset) and late activity with longer latency (> 350 ms after stimulus onset). In addition, the early activity was declined to the baseline level within ~200 ms during delay period, whereas the late activity lasted throughout the delay period and reached to the next stage of task (change color of the fixation point). Moreover, the early activity occurs primarily within the contralateral PFC of the visual stimulus, whereas the late activity occurs within both contralateral and ipsilateral PFC. While the early awareness related activity resembles the VAN activity in EEG studies (associating with p-consciousness), the late awareness related activity resembles the P3b activity (associating with a-consciousness). We are going to report these results in a separated paper soon.

I think the authors have to discuss the Gaillard et al PLOS Biology 2009 paper in much more detail. Gaillard et al also report a study related to conscious access contrasting unmasked and masked stimuli using iEEG. In this paper they also report ERP, time frequency and phase synchronization results (and even Granger causality). Because of the similarities in approach, I think it would be important to directly compare the results presented in that paper with results presented here and highlight the commonalities and discrepancies in the Discussion.

Thanks for reviewer’s comment. We have made additional analysis and detailed discussion accordingly. In addition, we also extended discussion with other relevant studies in the revised manuscript.

In lines 528-549,

‘Although one iEEG study reported awareness-specific PFC activation, the awareness-related activity started 300 ms after the onset of visual stimuli, which was ~100 ms later than the early activity in our study. Also, due to the limited number of electrodes in PFC (2 patients with 19 recording sites mostly in mesiofrontal and peri-insular regions), their experiments were restricted while exploring the awareness-related activity in PFC. In the present study, the number of recording sites (245) were much more than previous study and covered more areas in PFC. Our results further show earlier awareness-related activity (~ 200 ms after visual stimuli onset), including ERP, HFA and PLV. These awareness-related activity in PFC occurred even earlier (~150 ms after stimulus onset) for the salient stimulus trials (Fig. 3A\D and Fig. 4A\D, HA condition).

However, the proportions are much smaller than that reported by Gaillard et al, which peaked at ~60%. We think that one possibility for the difference may be due to the more sampled PFC subregions in present study and the uneven distribution of awareness-related activity in PFC. Meanwhile, we noticed that the peri-insula regions and middle frontal gyrus (MFG), which were similar with the regions reported by Gaillard et al, seemed to show more fraction of awarenessrelated sites than other subregions during the delay period (0-650 ms after stimulus onset). To test such possibility and make comparison with the study of Gaillard et al. we calculated the proportion of awareness-related site in peri-insula and MFG regions. We found although the proportion of awareness-related site was larger in peri-insula and MFG than in other subregions, it was much lower than the report of Gaillard et al. One alternative possibility for the difference between these two studies might be due to the more complex task in Gaillard et al. Nevertheless, we think these new results would contribute to our understanding of the neural mechanism underlying conscious perception, especially for the role of PFC.’ In lines 601-603:

‘The only human iEEG study reported that the phase synchronization of the beta band in the aware condition also occurred relatively late (> 300 ms) and mainly confined to posterior zones but not PFC.’

As for the Granger Causality analysis between PFC and occipital lobe, while the aim of this study focused mainly on PFC and there were few recoding sites in occipital lobe, we would like to do this analysis in later studies after we collect more data.

In the Gaillard paper they report a figure plotting the percentage of significant frontal electrodes across time (figure 4A) in which it can be seen that significant electrodes emerge after approximately 250 ms in PFC as well. It would be great if the authors could make a similar figure to compare results. In the current paper there are much more frontal electrode contacts than in the Gaillard paper, so that is interesting in itself.

Thanks reviewer for this constructive comment. We made similar analysis as Gaillard et al. and plotted the results in the figure bellow. As you can see, the awareness related sites started to emerge about 200 ms after visual stimulus onset according to both ERP and HG activity. The proportion of awareness related sites reached peak at ~14% (8% for HG) in 300-400ms. However, the proportions are much smaller than that reported by Gaillard et al, which peaked at ~60%. We think that one possibility for the difference may be due to the more sampled PFC subregions in present study and the uneven distribution of awareness-related activity in PFC. Meanwhile, we noticed that the peri-insula regions and middle frontal gyrus (MFG), which were similar with the regions reported by Gaillard et al, seemed to show more fraction of awareness-related sites than other subregions during the delay period (0-650 ms after stimulus onset). To test such possibility and make comparison with the study of Gaillard et al. we calculated the proportion of awareness-related site in peri-insula and MFG regions. We found although the proportion of awareness-related site was larger in peri-insula and MFG than in other subregions, it was much lower than the report of Gaillard et al. One alternative possibility for the difference between these two studies might be due to the more complex task in Gaillard et al.

We have added this figure and discussion to the revised manuscript as a new result (Figure 4E & S2 and lines 537-549).

**Author response image 1. sa4fig1:** Percentage of awareness-related sites in ERP and HG analysis. n, number of recording sites in PFC.

**Author response image 2. sa4fig2:** Percentage of awareness-related sites in ERP and HG analysis at parsopercularis and middle frontal gyrus (MFG). n, number of recording sites.

In my opinion, some of the most interesting results are not highlighted: the findings that subjectively unaware stimuli show increased activations in the prefrontal cortex as compared to stimulus absent trials (e.g., Figure 4D). Previous work has shown PFC activations to masked stimuli (e.g., van Gaal et al., J Neuroscience 2008, 2010; Lau and Passigngham J Neurosci 2007) as well as PFC activations to subjectively unaware stimuli (e.g., King, Pescetelli, and Dehaene, Neuron 2016) and this is a very nice illustration of that with methods having more detailed spatial precision. Although potentially interesting, I wonder about the objective detection performance of the stimuli in this task. So please report objective detection performance for the patients and the healthy subjects, using signal detection theoretic d'. This gives the reader an idea of how good subjects were in detecting the presence/absence of the gratings. Likely, this reveals far above chance detection performance and in that case I would interpret these findings as "PFC activation to stimuli indicated as subjectively unaware" and not unconscious stimuli. See Stein et al., Plos Biology 2021 for a direct comparison of subjectively and objectively unaware stimuli.

We gratefully appreciate for reviewer’s helpful and valuable comments. We do notice that the activity of PFC in subjectively unawareness condition (stimulus contrast near perceptual threshold) is significantly higher than stimulus absent condition. Such results, by using sEEG recordings with much higher spatial resolution than brain imaging and scalp EEG, support findings of previous studies (citations). Considering the question of neural correlation of unawareness processing is a hot and interesting topic, after carefully considering, we would like to report these results in a separate paper, rather than add these results in the current manuscript in order to avoid the distraction.

According to reviewer’s comment about the objective detection performance of the stimuli in our task, we analyzed the signal detection theoretic d’. The values of d’ in patients and healthy subjects are similar (1.81±0.27 in patients and 2.12±0.37 in healthy subjects). Such results indicate that the objective detection performance of subjects in our task is well above the chance level. Since our task merely measures the subjective awareness, we agree reviewer’s comment about the interpretation of our results as “PFC activation to stimuli indicated the subjective unawareness rather than objective unawareness”. We will emphasize this point in our next paper.

We have added the d prime in the MS (lines149-150).

In Figure 7 of the paper the authors want to make the case that the contrast does not differ between subjectively aware stimuli and subjectively unaware stimuli. However so far they've done the majority of their analyses across subjects, and for this analysis the authors only performed within-subject tests, which is not a fair comparison imo. Because several P values are very close to significance I anticipate that a test across subjects will clearly show that the contrast level of the subjectively aware stimuli is higher than of the subjectively unaware stimuli, at the group level. A solution to this would be to sub-select trials from one condition (NA) to match the contrast of the other condition (NU), and thereby create two conditions that are matched in contrast levels of the stimuli included. Then do all the analyses on the matched conditions.

Thank reviewer for the helpful comment. Regarding reviewer’s comment “However so far they've done the majority of their analyses across subjects, and for this analysis the authors only performed within-subject tests, which is not a fair comparison imo”, if we understand correctly, reviewer considered that it was fair if the analysis of neural activity in PFC was done across subjects but the stimulus contrast analysis between NA and NU was done individually. Actually, it is not the case. In neural activity analysis, the significant awareness-related sites were identified firstly in each individual subject (Fig. 3A and Fig 4A, and Methods), same as the analysis of stimulus contrast (see Methods). Only in the neural population activity analysis, the activity of awareness-related sites was pooled together and made further analysis.

To further evidence the awareness related activity in PFC is not highly correlated with stimulus contrast, we compared the activity difference between two different stimulus contrast conditions, i.e., stimulus contrast difference between high-contrast aware (HA) and NA conditions (large difference, ~14%), and between NA and NU conditions (slight difference, ~0.2%). The working hypothesis is that, if PFC activity is closely correlated with the contrast of stimulus contrast, we expect to see the activity difference between HA and NA conditions is much larger than that between NA and NU conditions. To test this hypothesis, we analyzed data of two patients in which the previous analysis showed significant or near significant difference of stimulus contrast between NA and NU conditions (Author response image 1, below, patient #2 and 1). The results (Author response image 1) show that the averaged activity difference (0-650 ms after visual stimulus onset) between HA and NA was similar as the averaged activity difference between NA and NU trials, even though the stimulus contrast difference was much larger between HA and NA conditions than between NA and NU conditions. Such results indicate that the awareness-related activity in PFC cannot be solely explained by the contrast difference between NA and NU conditions. Based on these results, we think that it is not necessary to perform the analysis as reviewer’s comment “A solution to this would be to sub-select trials from one condition (NA) to match the contrast of the other condition (NU), and thereby create two conditions that are matched in contrast levels of the stimuli included. Then do all the analyses on the matched conditions”. Another reason that impedes us to do this analysis is due to the limited trial numbers in our dataset.

**Author response image 3. sa4fig3:** Relationship between stimulus contract and PFC activity. X axis represents the stimulus contrast difference between two paired conditions, i.e., aware versus unaware in near perceptual threshold conditions (NA – NU, red dots); aware in high contrast condition versus aware in near perceptual threshold condition (HA – NA, blue dots). Y axis represents the activity difference between paired stimulus conditions. The results show that activity difference is similar between two paired conditions regardless the remarkable contrast difference between two paired conditions. Such results indicate that the greater activity in NA trials than in NU trials (Fig. xx-xx) could not be interpreted by the slight difference in stimulus contrast between NA and NU trials.

Related, Figure 7B is confusing and the results are puzzling. Why is there such a strong below chance decoding on the diagonal? (also even before stimulus onset) Please clarify the goal and approach of this analysis and also discuss/explain better what they mean.

We have withdrawn Figure7B for the confusing decoding results on the diagonal.

I was somewhat surprised by several statements in the paper and it felt that the authors may not be aware of several intricacies in the field of consciousness. For example, a statement like the following "Consciousness, as a high-level cognitive function of the brain, should have some similar effects as other cognitive functions on behavior (for example, saccadic reaction time). With this question in mind, we carefully searched the literature about the relationship between consciousness and behavior; surprisingly, we failed to find any relevant literature." This is rather problematic for at least two reasons. First, not everyone would agree that consciousness is a highlevel cognitive function and second there are many papers arguing for a certain relationship between consciousness and behavior (Dehaene and Naccache, 2001 Cognition; van Gaal et al., 2012, Frontiers in Neuroscience; Block 1995, BBS; Lamme, Frontiers in Psychology, 2020; Seth, 2008 and many more). Further, the explanation for the reaction time differences in this specific case is likely related to the fact that subjects' confidence in that decision is much higher in the aware trials than in the unaware trials, hence the speeded response for the first. This is a phenomenon that is often observed if one explores the "confidence literature". Although the authors have not measured confidence I would not make too much out of this RT difference.

We agree that and modified accordingly in lines 492-507.

‘An alternative interpretation for RT difference between aware and unaware condition in our study, i.e., reflecting task-strategies used by subjects/patients to remember the response mapping rules between the perception and the color cue (e.g., if the YES+GREEN=RIGHT and YES+RED=LEFT rules were held in memory, while the NO mappings were inferred secondarily rather than being actively held in memory).

Another possibility is that the reaction time is strongly modulated by the confident level, which has been described in previous studies(Broggin et al., 2012; Marzi et al., 2006). However, in previous studies, the confident levels were usually induced by presenting stimulus with different physical property, such as spatial frequency, eccentricity and contrast. However, the dependence of visual process on the salience of visual stimulus confounds with the effect of visual awareness on the reaction time of responsive movements, which is hard to attribute the shorter reaction time in more salient condition purely to visual awareness. In contrast, we create a condition (near aware threshold) in the present study, in which the saliency (contrast) of visual stimulus is very similar in both aware and unaware conditions in order to eliminate the influence of stimulus saliency in reaction time. We think that the difference in reaction time in our study is mainly due to the modulation of awareness state, which was not reported previously.’

I would be interested in a lateralized analysis, in which the authors compare the PFC responses and connectivity profiles using PLV as a factor of stimulus location (thus comparing electrodes contralateral to the presented stimulus and electrodes ipsilateral to the presented stimulus). If possible this may give interesting insights in the mechanism of global ignition (global broadcasting), supposing that for contralateral electrodes information does not have to cross from one hemisphere to another, whereas for ipsilateral electrodes that is the case (which may take time). Gaillard et al refer to this issue as well in their paper, and this issue is sometimes discussed regarding to Global workspace theory. This would add novelty to the findings of the paper in my opinion.

We gratefully appreciate reviewer’s helpful and available suggestions. We have made the analysis accordingly. We find that the awareness-related ERP activation in PFC occurs earlier only in the contralateral PFC with latency about 200 ms and then occurs in both contralateral and ipsilateral PFC about 100 ms later. In addition, the magnitude of awareness-related activity is stronger in the contralateral PFC than in ipsilateral PFC during the early phase (200-400 ms), then the activity becomes similar between contralateral and ipsilateral PFC. Moreover, the awareness related HG activity only appears in the contralateral PFC. Such results show the spatiotemporal characteristics of visual awareness related activity between two hemispheres. We are going to report these results in a separate paper soon.

**Reviewer #3 (Recommendations For The Authors):**
Some of the font sizes in the figures are too small.

We have modified accordingly.

To me, the abbreviations are confusing, (NA/NU etc). I would try to come up with easier ones or just not use abbreviations.

We have modified accordingly and try to avoid to use the abbreviations.

The data/scripts availability statement states "available upon reasonable request". I would suggest that the authors make the data openly available when possible, and I believe eLife requires that as well.

Thanks for reviewer’s suggestions. Due to several ongoing studies based on this dataset, we would like to open our data after complete these studies if there is no restriction from national policy.